

1 **The Influence of a Prolonged Meteorological Drought on the Catchment**

2 **Water Storage Capacity: A Hydrological Model Perspective**

3 **Zhengke Pan[a,b,c], Pan Liu[a,b,*], Chongyu Xu[c,a,b], Lei Cheng[a,b], Jing Tian[a,b], Shujie Cheng[a,b], Kang Xie[a,b]**

[a]State Key Laboratory of Water Resources and Hydropower Engineering Science, Wuhan University,
Wuhan 430072, China
[b]Hubei Provincial Key Lab of Water System Science for Sponge City Construction, Wuhan University,
Wuhan, Hubei, China
[c]Department of Geosciences, University of Oslo, P O Box 1047 Blindern, Oslo N-0316, Norway
*Corresponding author. Email: liupan@whu.edu.cn; Tel: +86-27-68775788; Fax: +86-27-68775788





## Abstract

Understanding the propagation process of prolonged meteorological droughts (i.e., decade) helps solve the problem of increasing water scarcity around the world. Historical literature studied the propagation between different drought types (e.g., from meteorological to hydrological drought) with mainly statistical approaches, however, little attention has been paid to the causality between the meteorological drought with potential changes in the Catchment Water Storage Capacity (CWSC) where the latter plays a critical role in catchment response behavior to former. This study used the temporal variation in the estimated value of a model parameter that denotes the CWSC in its model structure to reflect the potential changes in real CWSC. The most likely change points of the CWSC were determined based on the Bayesian change point analysis. Also, the possible association and linkage between the shift in the CWSC and the time-lag of the catchment (i.e., time-lag between the onset of the drought and the change point) with multiple catchment properties and climate characteristics have been studied. Catchments from southeastern Australia were used as a study area to verify the effectiveness of the proposed approach. Results indicated that (1) in 62.7% of the catchments, the sustained drought causes significant shifts in the CWSC. The shift led to the opposite response in two subsets of catchments, i.e., 48.2% of catchments had lower runoff generation rates for a given rainfall while 14.5% of catchments had higher runoff generation rate. (2) Catchments with larger elevation and slope, lower forest coverage of Evergreen Broadleaf Forest are more likely to have an increase in the CWSC during a chronic drought while smaller catchments with lower elevation, lower coverage of the Evergreen Broadleaf Forest are more likely to have a decrease in the CWSC. (3) The changed catchments were not equally susceptible to the pressure due to persistent meteorological drought. Catchments with a lower proportion of Evergreen Broadleaf Forest usually have longer time-lag and are more resilient. This study improves our understanding of possible changes in CWSC induced by a prolonged meteorological drought, which will help improve our ability to simulate the hydrological system.

**Keywords:** Catchment water storage capacity; GR4J hydrological model; Temporal Variation; Meteorological drought; Catchment response; Resilience



## 1. Introduction

Drought is one of the most damaging environmental disasters and has significant environmental, economic, and social impacts around the world (Zhao and Running, 2010). It not only affects the balance of aquatic and terrestrial ecosystems, but also many economic and social sectors including agriculture yield, industrial production and urban water supply (Mishra and Singh, 2010). Furthermore, recent literature indicates that the improved probabilities of the extreme events have been projected in many parts around the world because of the increasing anthropogenic interference and global climate change, which implies the more severe droughts in the future (Dai, 2012;Dai, 2011;Pan et al., 2019b). Unlike other natural hazards, e.g., flood, cyclonic storms and earthquakes, the drought may have a much longer duration.

The droughts are generally classified into four categories (Mishra and Singh, 2010): meteorological, agricultural, hydrological, and socio-economic droughts. All droughts start as meteorological droughts caused by precipitation shortage. Then a prolonged meteorological drought might result in a hydrological drought that is represented by the deficits in the surface or subsurface water supply (e.g., streamflow, groundwater, reservoir, and lake storage). An agricultural drought, a period with declining soil moisture and consequent crop failure without any reference to surface water resources, is usually the combined effects of meteorological and hydrological droughts. A socio-economic drought is associated with the failure of water resources systems to meet water demands as a result of a weather-related shortfall in the water supply. Recently, many studies have been carried out on the links between different drought types, e.g., the link between the meteorological and hydrological droughts (Haslinger et al., 2014;Huang et al., 2017;Lopez-Moreno et al., 2013;Lorenzo-Lacruz et al., 2013;Van Loon et al., 2015;Wu et al., 2017), the link between meteorological and agricultural droughts (Huang et al., 2015;Wu et al., 2016), and the link between meteorological and socio-economic droughts (Zhao et al., 2019). However, little attention has been paid to the causality between the meteorological drought with changes in catchment properties where the latter plays a critical role in the transference of different drought types.

Whether a sustained shift in precipitation (e.g., a prolonged meteorological drought) can trigger a change in catchment properties is important for understanding the mechanism of catchment response and hydrological projections under change (Schindler and Hilborn, 2015). For example, because of the significant decline in annual rainfall in the late 1960s, a shift from perennial to ephemeral streams and a decline in the runoff coefficient (runoff/rainfall) since the 1970s have been observed in Western Australia (Petrone et al., 2010). Furthermore, Saft et al. (2015) indicated the shift in annual rainfall-runoff relationship in southeastern Australia during the Millennium drought (1997-2009). The possible mechanisms may be that the drought-induced persistent change in groundwater level (Van Lanen et al., 2013), catchment soil condition (Descroix et al., 2009), vegetation (Adams et al., 2012) and soil moisture (Grayson et al., 1997). Furthermore, a new hydrologic regime has developed in the study area with important implications for future surface water supply.

One of the most important attributes of a catchment is the ability to store water and to release it later, which is known as the catchment water storage capacity (CWSC). The storage serves as a buffer for climate variability, meteorological extremes, and sustain vegetation during the drought





period, while the value of CWSC is a vital index to quantify and compare the maximum water
volume of different catchments. The detailed definition (McNamara et al., 2011) of CWSC is in an
unregulated and unimpaired catchment, the water storage capacity is defined as the maximum
water volume that a catchment can hold after rainfall events. It refers to the part of effective rainfall
that does not develop into the surface flow, and it is the sum of soil water storage capacity,
vegetation intercept, and snowpack. Recently, there are two main methods to derive the CWSC
value, i.e., water balanced method and hydrological modeling method. For the former method,
$V(T) = V_0 + \Delta t \sum_{t=1}^{T}(P_t - Q_t - E_t)$, where $V(T)$ is the storage at time step $t = T$, $\Delta t$ is the
interval between two contiguous time steps, $V_0$ is the storage at time $t = 0$, $P_t$, $Q_t$ and $E_t$
refer to the precipitation (mm), streamflow (mm) and evapotranspiration (mm) at time step $t$,
respectively. Thus, the CWSC is denoted as the difference between minimum and maximum of the
computed annual storage volumes over the observation period. For the latter method, the CWSC
is estimated through the calibration of hydrological model parameters (that denote the catchment
water storage capacity within the model structure) with the time series of precipitation,
evapotranspiration and streamflow as well as certain objective functions (e.g., minimize the errors
between the streamflow observations and the simulated results based on the estimated
parameters) and the inference methods (e.g., SCEM-UA algorithm by Vrugt et al. (2003)). Generally,
the latter has the advantage of quantifying the contribution of snow, soil and groundwater storage
(Staudinger et al., 2017). For example, Deng et al. (2016) calibrated the time-varying parameters
of a two-parameter monthly water balance model with a case study in Wudinghe catchment of
China and found one of the model parameters that denotes the CWSC experienced a significant
upward trend during the historical period from 1958 to 2000. Pan et al., (2019b) calibrated the
GR4J hydrological model with time-varying parameters in three catchments of southeastern
Australia, and found that the spatial coherence of adjacent catchments helps to reduce the
estimation uncertainty of CWSC and improve the model prediction performance. In addition,
because of the resilience of catchment, the shift in CWSC might occur as a delayed step-change.
However, no study has been made on the application of hydrological modeling methods to explore
the impacts of sustained meteorological drought on the catchment water storage capacity.
The objectives of this study, therefore, are: (1) to verify whether a sustained meteorological
drought would result in a significant change in CWSC; if so, to explore the possible change points
(the time points that the value of the CWSC experienced an abrupt variation), change direction
(whether the value of the CWSC has an upward or downward change after the change point) and
change magnitude (the difference between the values of CWSC during the periods before and after
the change point); (2) to analyze which catchment properties and climate characteristics are more
promising to be related to the shift in CWSC; and (3) to explore the possible association between
catchment properties and climate characteristics with the time-lag between the onset of the
meteorological drought and the change point.
The remainder of this study is organized as follows. Section 2 presents the study area and research
data. Section 3 illustrates the methodology to explore the questions mentioned above. Section 4
provides the results of catchments with significant and non-significant changes in CWSC as well as
catchments with different change directions due to a prolonged meteorological drought, illustrates
the association results between the shift in CWSC and time-lag with the potential variables (include
catchment properties and climate characteristics). Section 5 provides discussions of our results.



The conclusions have been made in section 6.

## 2. Study area and data

### 2.1. Study area

Analyses of this paper are based on daily rainfall, potential evapotranspiration, runoff time series,
and catchment attributes from southeastern Australia. The catchments were checked to be free
from major anthropogenic disturbances during the measurement history (Zhang et al., 2013).
Southeastern Australia has gone through nearly a decade of meteorological drought that was
approximately from 1996 to 2009. This drought has resulted in large impacts on the economy,
culture, political and ecosystem development of southeastern Australia, the most densely
populated part of Australia. The study catchments are situated in this region, extending from
southern Queensland, southern New South Wales, and whole Victoria. A map of the study area
with geolocation of the study catchments in southeastern Australia is demonstrated in Figure 1.
The study catchments exhibit a broad variety of climatic conditions, soils, land use, vegetation, and
hydrological regimes. Generally, the study catchments have much more rainfall during the spring
and winter seasons than the summer and autumn seasons. In most of the study catchments, there
is no snowmelt; even if the snowmelt appears in an individual catchment, it does not have much
effect on the local hydrological system because the mean elevation of these catchments is around
584 m. It should be mentioned that the mean elevation of these catchments is much lower than
the seasonal snow line (1500 m) in this area (Saft et al., 2015).

### 2.2. Research data

The following data have been used in this study: (1) climate inputs, include daily rainfall and daily
potential evapotranspiration; (2) daily runoff at catchment outlet; and (4) catchment attributes,
include catchment area, mean elevation and so on. The data were obtained from the Australian
Water Resources Assessment (AWRA) system, which has been served as a standard publicly
available national dataset for hydrological model evaluation
(https://publications.csiro.au/rpr/pub?pid=csiro:EP113194, Zhang et al. (2013)). For all
catchments, there is no missing data in the rainfall and potential evapotranspiration data while the
runoff data in some catchments are missing.
398 catchments from southeastern Australia were selected from the original data set, both of these
catchments were not regulated, and with insignificant effects of human activities during the
observation period with the catchment area from 50 to 17000 $km^2$. Records of these catchments
are both ranged from January 1, 1976, to December 31, 2011. For the initial data set of 398
catchments from southeastern Australia, a set of 125 catchments were excluded in advance
because the completeness of daily streamflow data in these catchments is less than 80 percent.
The remaining 273 catchments are used for identification of meteorological droughts. Finally, 145
catchments within the subset were identified with a long-term meteorological drought (see
sections 3.1 and 4.1) and analyzed further. The attributes of the 145 catchments are summarized
in Table S1 (Supplement material).



## 3. Methodology

This section presents the methodology of this study, including (i) identification of catchments with a long-term meteorological drought (section 3.1); (ii) derivation of change in CWSC on account of drought based on hydrological modeling method (section 3.2), includes an introduction of the hydrological model adopted, the likelihood function, model parameter estimation method, and the identification method for the change points of CWSC; (iii) potential variables that might be associated with the potential changes in CWSC (section 3.3).

### 3.1. Identification of meteorological drought

The method proposed by Saft et al. (2015) was introduced in this study to define the meteorological drought period (also known as dry period), which was only based on the annual rainfall data. Saft et al. (2015) have examined several algorithms for the identification of dry period with the consideration of different combinations of the dry period anomaly (i.e., the percentage variation between the annual rainfall during the dry period and the average of whole time series), length of the dry period and various boundary conditions, the delineation results of the dry period by one of the algorithms illustrated the lowest dependency on the algorithm itself, and were robust to different algorithms (Pan et al., 2019b). The process of this best identification method is generalized as follows:

Firstly, the anomaly was calculated as the percentage variation between the annual rainfall data and the annual mean value, and the anomaly was smoothed with a 3-year moving window. It should be mentioned that smoothing was applied to avoid single wetter years that interrupts a long dry period and identify all period of consecutive smoothed negative anomalies. Secondly, the start year of the dry period was defined as the start of the first 3 continuous years of the negative anomaly period based on the unsmoothed anomaly data. Similarly, the end year of the dry period was determined from the last negative 3-year anomaly series based on the unsmoothed anomaly data. The end year was defined as the last year of this 3 year series unless (i) there was a year with a positive anomaly that was larger than 15% of the mean, in which case the end of dry period was determined as the year prior to that year, (ii) the last 2 years had a bit positive anomalies (but each was smaller than 15% of the mean), in which case the end year was determined as the first year of positive anomaly. Two additional rules were set to ensure a sufficiently long and severe dry period: the length of dry period should longer than 6 years and the mean dry years anomaly should be smaller than 5%. In addition, the remaining part in the observation history (except the dry period) was determined as the non-dry period.

The selected algorithm has been verified as a rigorous method for processing the autocorrelation in regression residuals and testing the global significance. Furthermore, we have the same study region, i.e., catchments in southeastern Australia (but our data sources are different). Thus, the method proposed by Saft et al. (2015) was introduced in this study to define the meteorological drought period. A more detailed process of the identification method of dry period can be obtained in research by Saft et al. (2015) and our previous study (Pan et al., 2019b).



## 3.2. Derivation of the catchment response to drought


3.2.1 Hydrological model
The conceptual rainfall-runoff model, i.e., the GR4J (modèle du Génie Rural à 4 paramètres
Journalier) hydrological model, was used to examine the proposed method (Perrin et al., 2003).
Previous studies showed that the GR4J model had comparable simulation and prediction
performance with other hydrological models with more model parameters (Pan et al., 2019a;Pan
et al., 2019b;Westra et al., 2014). The GR4J model comprised four parameters: $\theta_1$ represents the
catchment water storage capacity (mm); $\theta_2$ denotes the coefficient of groundwater exchange
(mm); $\theta_3$ represents the 1-day-ahead maximum capacity of the routing store (mm); and
$\theta_4$ denotes the time base of the unit hydrograph (days). Previous studies (Pan et al., 2019a;Pan et
al., 2019b;Perrin et al., 2003;Westra et al., 2014) showed that $\theta_1$, which denotes the catchment
water storage capacity, is the most sensitive parameter in the structure of the GR4J model.
In the GR4J model structure, the first operation is to subtract evapotranspiration from the original
rainfall to determine the net rainfall or net evapotranspiration. The net rainfall is then divided into
a surface flow and a water production storage of catchment, where $\theta_1$ is the maximum capacity
of the production store of the catchment. The total runoff includes two flow components
(underground flow and the surface runoff) is then divided into fast and slow routing processes that
are defined by two unit hydrographs. 90% of the total runoff is routed by the slow unit hydrograph
and then a non-linear routing store, while the remaining 10% of the runoff is propagated through
the fast routing processes. Both unit hydrographs depend on the same parameter $\theta_4$ expressed
in days. In addition, a groundwater exchange term that acts on both flow components is calculated
based on parameter $\theta_2$ and $\theta_3$. More details about the Gr4J model can be found in Perrin et al.

237 (2003).

The real CWSC values are hard to derive out based on available data and attributes of catchments.
However, the hydrological model provides a new perspective for reflecting the potential variations
of the CWSC, that is, the utilization of specific model parameter(s) that represent(s) CWSC, such as
parameter $\theta_1$ in the GR4J model. Figure 2 presents an example to illustrate the impacts of the
shift in the value of $\theta_1$ on the model simulation results.
Thus, in this study, we use the magnitude of the shift in estimated $\theta_1$ between periods before and
after the change point to represent the change in CWSC. In addition, we assume that the other
model parameters $\theta_2$, $\theta_3$, and $\theta_4$ are kept constant during the periods before and after the
change point, and the shift of CWSC happens on the potential change point. The constant
assumption of parameters $\theta_2$, $\theta_3$, and $\theta_4$ is a common assumption, which has been made in
many previous literatures, such as Westra et al. (2014), Pan et al. (2019a) and Pan et al. (2019b).
3.2.3 Likelihood function, parameter estimation, and inference
(1) likelihood function
For catchment *c*, the likelihood function in this study was adopted from Thiemann et al. (2001),
which is written as:



$$p_c\left(\vec{\theta}(c)\Big|\xi(c),q(c),\tau\right) \propto \left[\frac{\omega(\tau)}{\sigma}\right]^T \exp\left[-c(\tau)\sum_{t=1}^{T}\left|\frac{e_t\left(\vec{\theta}(c)\right)}{\sigma}\right|^{2/(1+\tau)}\right] \cdot p\left(\vec{\theta}(c)\right) \quad (1)$$

where
$$\omega(\tau)=\frac{\left\{\Gamma\left[3(1+\tau)/2\right]\right\}^{1/2}}{(1+\tau)\left\{\left\{\Gamma\left[(1+\tau)/2\right]\right\}^{3/2}\right\}}, c(\tau)=\left\{\frac{\Gamma\left[3(1+\tau)/2\right]}{\Gamma\left[(1+\tau)/2\right]}\right\}^{1/(1+\tau)} \quad (2)$$

where $p$ refers to the likelihood probability. $\vec{\theta}(c)=\{\theta_1,\theta_2,\theta_3,\theta_4\}$ and $\Gamma(.)$ indicates the
gamma function. $T$ refers to the number of time step $t$; $c$ is the index of catchments; $q$ denotes
observations of streamflow; $\xi$ are the climate inputs of the hydrological model, including
precipitation (P) and potential evapotranspiration (PET); $e_t$ signifies the residual error at time step
$t$; and $\tau$ refers to the residual error model type (Pan et al., 2019a;Thiemann et al., 2001). When
the model type of residual error is verified, parameter $\omega$ and $c$ are unchanged values. The
Gaussian distribution was used to denote the residual error model in this study, thus $\tau=0$ was
verified. Therefore, $\omega(\tau=0)$ and $c(\tau=0)$ are confirmed as: $\omega(\tau=0)=\frac{\{\Gamma(3/2)\}^{1/2}}{\{\Gamma(1/2)\}^{3/2}}=$
$0.3989, \omega(\tau=0)=\frac{\Gamma(3/2)}{\Gamma(1/2)}=0.5$, respectively. Additionally, for all unknown quantities, uniform
distributions are used as their prior distributions.

266        (2) Inference

The estimation of the posterior distributions of all unknown quantities is based on the Shuffled
Complex Evolution Metropolis (SCEM-UA) sampling method (Vrugt et al., 2003). The Gelman-Rubin
convergence value (Gelman et al., 2013) was used as the evaluation criterion of model convergence,
with its value should be smaller than the threshold 1.2. The prior ranges of all model parameters
have been given in Table 1.
3.2.4 Bayesian change point analysis
In this part, the Bayesian change point analysis was introduced to find out the possible change
points of the CWSC. Change point of the CWSC denotes the time point that the estimated values
of the CWSC between two periods (before and after this point) were significantly different. Each
change point was then illustrated by a likelihood probability. The time point with the maximum
likelihood probability among all potential options was regard as the final change point of the
catchment.
The Bayesian change point is one of the simplest and most effective methods to analyze the change
point problem (Cahill et al., 2015;Carlin et al., 1992). The detailed process follows:
At first, assume unknown value $k$ as the potential change point of CWSC and its value is taken from
$\Theta=\{1,2,\dots,T\}$. Thus, $k=T$ is interpreted as "no change". Assuming the rainfall-runoff
relationship $Y$ before the change point $k$ is


$$Y_i(c) \sim p_c\left(\vec{\theta}(c)\middle|\xi_i(c), q_i(c), \tau\right), i = 1, \ldots, k-1.$$
(3)

Therefore, for all possible $k$, corresponding values of calibrating parameters $\vec{\theta}(c)$ can be obtained
through the derivation of the likelihood function with the SCEM-UA algorithm. Obviously, there
were four unknown quantities (i.e., four model parameters) that need to solve during this process.
Similarly, the rainfall-runoff relationship after the change point $k$ is assumed as
$$Y_i(c) \sim p'_c\left(\overrightarrow{\theta'}(c)\middle|\xi_i(c), q_i(c), \tau\right), i = k, \ldots, T.$$
(4)

where $\overrightarrow{\theta'}(c) = \{\theta'_1, \theta_2, \theta_3, \theta_4\}$. Parameters $\theta_2, \theta_3$ and $\theta_4$ are predefined and taken from the
calibrating results of the previous process. Thus, only $\theta'_1$ would be calibrated in this process. The
association between $p_c$ and $p'_c$ is that the posterior distributions of $\theta_2, \theta_3, \theta_4$ are the same.
Therefore, for every $k$, the likelihood function $L(Y; k)$ becomes
$$L(Y; k) = p_c\left(\vec{\theta}(c)\right) \cdot p'_c\left(\overrightarrow{\theta'}(c)\right)$$
(5)

The Bayesian perspective is added by placing a prior density $\varsigma(k)$ on $\Theta$ whence the posterior
density of $k|Y$
$$p(k|Y) = \frac{L(Y; k)\varsigma(k)}{\sum_{j=1}^{n} L(Y; j)\varsigma(j)}$$
(6)

Therefore, the final Change pint is recognized as the point that has the largest $p(k|Y)$. It should
be noted that every catchment would get a "potential change point" during the calculation process.
However, these "potential change points" would be evaluated to judge whether there is a
significant shift in the estimated $\theta_1(\theta'_1)$ between these two periods, i.e., before $k$ and after $k$. In
order to speed up the calculation process, the time interval between two adjacent change point is
set as 30 days.
3.2.5 Criteria in identifying catchments with a significant change in $\theta_1$
In order to derive the catchments with significant changes in CWSC. Four evaluation criteria have
been used in this study.
(1) **The minimum NSE requirement**: The Nash-Sutcliffe efficiency coefficient (NSE) values in
two periods should be larger than 0.6. The reason for setting this requirement was to ensure the
simulation results by the adopted model (the GR4J model) was reasonable for the simulation of
the hydrological cycle in a catchment, thus, the estimated model parameter can truly reflect the
CWSC of this catchment.
(2) **The minimum requirement of significant change:** The simulated values of the parameter
$\theta_1$ between the two periods should be more than ±20%. In other words, only the catchments with
more than ±20% changes in $\theta_1$ would be recognized as significantly changed. It should be noted
that the value of ±20% was an experienced value, actually, we have tried several other significant





levels (such as ±5% and ±10%), and found that the value of ±20% can maximally exclude the
negative impacts by the heterogeneity of the available parameter sets.
(3) **The requirement for maximum performance degradation:** NSE performance degradation
between the two periods should be no more than 20%.
(4) **The requirement for the robustness of results:** The initial conditions (i.e., the initial value
of all unknown quantities which might have impacts on the final results) for the calculation would
change three times, only the catchments that were identified as significantly changed in each time
would be identified as the final changing items.

## 324   3.3. Response time of catchment

In those catchments with a significant change in CWSC, a time-lag would usually exist between the
onset of the meteorological drought and the change point because of the catchment resilience.
For example, Van Lanen et al. (2013) and Huang et al. (2017) indicated that the groundwater
maintained the runoff during a brief drought period thus acted as a cushion to the spread of
meteorological drought to hydrological drought. However, the interactions between the surface
water and groundwater would be gradually reduced because of the falling groundwater levels if
the drought conditions persist for several years and even decades. The shift between the
connected situation and disconnected condition usually takes some time, and occurs as a delayed
step-change, which is also known as the time-lag of the catchment. Furthermore, the vegetation,
catchment soil moisture may also have an impact on the response of catchments to meteorological
drought.

## 336   3.4. Potential factors

Since the processes potentially responsible for the shift in CWSC are not directly measured, thus
some measurable proxies are used to explore the potential factors and possible associations
between these potential factors with the changes in CWSC and the length of the time-lag of the
catchment. Thus, 33 variables, including catchment physical properties and climate characteristics,
were employed. It should be noted that because of the limitation of available data of catchment
attributes, only unique catchment properties are employed. However, each climate variable
includes four values, i.e., the values of climate characteristics during the periods before and after
the change point, and the variation and percent variation of the climate characteristics between
the aforementioned two values. For example, four values of the average of daily runoff are
considered, i.e., the average values of daily rainfall during the period before the change point
$AR_{before}$, the average values of daily rainfall during the period after the change point $AR_{after}$, the
variation and percent variation of the average of daily rthe unoff between $AR_{before}$ and $AR_{after}$.
Table 2 summarizes the potential factors included in the following analysis. The employed climate
variables can be divided into four aspects, i.e., daily (Y1-Y4), monthly (Y5-Y7), seasonal (Y8-Y16)
and annual scale variables (Y17-Y24). Noted that the base flows of catchments were calculated
based on the Lyne-Hollick method (Lyne and Hollick, 1979).



## 4. Results

### 4.1. Catchments with a long-term meteorological drought

125 catchments in southeastern Australia were excluded from the original data set (total 398 catchments) because these catchments lacked a long enough data series during their streamflow measurement history, that is, the completeness of daily streamflow data is less than 80 percent. Furthermore, 145 catchments from the filtered 273 catchments have been identified with one long-term meteorological drought during its observation history according to the identification method mentioned in section 3.1. It should be mentioned that the catchments with more than one long-term drought periods in its measurement history were not included in order to exclude the unnecessary impact on the subsequent evaluation of shift in CWSC due to sustained drought. For most catchments, the long-term meteorological drought started around 2000 and then ended around 2009. The drought length of all these catchments was longer than 7 years. During the period, a larger than 5 percentage decrease has been identified in the annual rainfall of all those catchments.

### 4.2. Catchments with significant and non-significant change

The Bayesian change point test was applied to the 145 catchments that have been identified with a long-term meteorological drought (Table S1). Based on the evaluation criteria mentioned in section 3.2.5, it was found that 83 of the 145 catchments satisfied the requirements for minimum NSE performance and maximum performance degradation. The following analysis was based on these 83 catchments.

In 52 out of 83 (62.7%) catchments, the estimated value of the model parameter $\theta_1$ was detected to have a significant change after the onset of the meteorological drought, indicating the potential changes of CWSC in these catchments. These catchments satisfied all criteria mentioned in section 3.2.5. More clearly, the median estimated value of parameter $\theta_1$ during the period after the change point was significantly different with that value during the period before the change point; the median NSE performances of both parameter sets in two periods were larger than 0.6; furthermore, the performance degradation between simulated results before and after the change point was no more than 20 percent. Meanwhile, the remaining 31 (37.3%) catchments had no significant changes after the onset of the drought.

Figure 3 illustrates the results of three examples, including the posterior distributions of $\theta_1$ during the periods before and after the change point (left three columns), the posterior probability in the likely change points (median three columns) and the corresponding NSE performance (right three columns). Clearly, there was a time-lag between the onset of the meteorological drought and the shift in CWSC. In catchment 215002, a significant upward change in $\theta_1$ was detected after the change point. It means that after the change point the CWSC was larger than expected based on the calibrated results in two periods. As for the posterior probability, it was found that there was a probability of 48.0% that the change point would locate in the range of [July in 2001, December in 2001], and a probability of 76.0% that the change point would locate in the range of [February in





2001, December in 2001]. In catchment 401203, the white box that refers to the mean estimate of
$\theta_1$, was shifted downward significantly, indicating the potential decreased change in CWSC of this
catchment. The likely change point has a probability of 39.9% to occur within the period of [January
in 2003, July in 2003]. Finally, catchment 410061 experienced an upward but no significant change
in $\theta_1$ when comparing the results from two periods that separated by the "most likely change
point". Since the shift in $\theta_1$ was not significant in this catchment, the change point did not exist
here statistically. Thus, under sustained rainfall reduction, the CWSC of different catchments might
experience absolutely diverse changes. The possible reasons may lie in the diverse catchment
properties and climate characteristics (see sections 4.4 and 4.5).

## 400    4.3. Direction and magnitude of the shift in CWSC

In the case of the direction of shift in CWSC, a significant increase in the estimated $\theta_1$ has been
found in 40 out of 83 catchments. Since the significant decrease in rainfall has been found during
the prolonged drought, these catchments were expected to experience a downward trend with a
similar magnitude of reduction in the runoff generation. However, the increase of CWSC means
that the drought might result in lower runoff generation rates for similar amounts of rainfall during
the drought period. Thus, in the following years with reduced rainfall, lower runoff due to the
reduced rainfall could be expected, furthermore, the even less runoff than historical records may
occur because of the significant increase in CWSC. Another 12 catchments had a significant
downward shift in the CWSC. The decrease of CWSC indicates that the meteorological drought
might result in higher runoff generation rates for a similar amount of rainfall than previous records.
These catchments had the lower capacity to hold available water and their ecosystems might suffer
more frequent and more severe extreme events, e.g., droughts and floods. In addition, the
remaining 31 catchments were divided into two parts further, i.e., 18 catchments with a slight (non-
significant) increase in the CWSC and 13 catchments with a slight (non-significant) decrease in the
CWSC. As for the geographical distribution of the catchments with significant and non-significant
changes in the CWSC, Figure 4 illustrates that there is some tendency for clustering, e.g., (1) for
majority, adjacent catchments tend to have same change directions; (2) catchments in
southwestern and southern Victoria experienced different levels of increase in the CWSC, while (3)
in northeastern Victoria, the majority of catchments had a decrease in the CWSC. However, the
geographical distribution of catchments with significant and non-significant changes in $\theta_1$
showed no obvious geolocation clustering phenomenon. The magnitude of change in the CWSC
was analyzed based on the shift in the estimated value of $\theta_1$ of two periods. Figure 5 presents the
statistical histograms of catchments with different degrees of the shift in $\theta_1$. It should be noted
that both catchments with significant changes and non-significant changes have been plotted
together. The fitted curves in Figure 5 (a) and (b) are both positive-biased since that there was a
larger number of catchments with an increasing trend in the CWSC rather than those with
decreased trend detected. The distribution of the catchments illustrates that the majority of
catchments have the [-50, 100] percent change (based on the period before the change point) or
[-100, 200] absolute change in the estimated value of $\theta_1$.





### 4.4. Factors for shifts in the CWSC

In this part, we investigate whether changes in the CWSC are associated with particular catchment properties or climate characteristics. In other words, are catchments with certain catchment properties or climate characteristics easier to trigger the potential shift in its CWSC?

4.4.1 Factors for the magnitude of shifts in the CWSC

Two groups of catchments were established according to the significance level of the shift in $\theta_1$ after the onset of the long-term drought. Specifically, one group is composed of catchments with a significant change in the estimation of parameter $\theta_1$ between two periods. The other group is composed of catchments that only had a non-significant change in parameter $\theta_1$ between two periods. As shown in Figure 6 (left two columns), the catchment properties of two groups of catchments have been presented. Significant changes in the CSWC were likely to occur in catchments with smaller catchment areas, lower elevation and its difference, less slope, lower Available soil Water Holding Capacity (AWHC) in sub soil, smaller saturated hydraulic conductivity (Ks) in top soil. The forest coverage percentage and the AWHC in top soil are not significantly different between the two groups of catchments. It should be noted that some of those catchment properties might be somewhat related. Thus, the Pearson Correlation Coefficient (PCC) has been used to explore the potential relationship between the change in parameter $\theta_1$ with the catchment properties as well as the connection between different catchment properties. Figure 7 illustrates a low degree (PCC $\in [\pm 0.3, \pm 0.5]$) of association between the (%) shift in parameter $\theta_1$ and these catchment properties for two groups of catchments.

4.4.2 Factors for the direction of shifts in the CWSC

Two subsets of catchments were established according to the change direction of the estimated $\theta_1$. Specifically, one subset is composed of catchments with a significant upward change in the estimated $\theta_1$ while another subset is composed of catchments that experienced a significant downward shift in the estimated $\theta_1$. As shown in Figure 6 (right two columns), catchments with a significant upward change in estimate $\theta_1$ had a smaller catchment area, lower elevation, and its difference, less slope, lower AWHC in sub soil, smaller saturated hydraulic conductivity (Ks) in top soil. The forest coverage percentage, the AWHC in top soil and the length of the time-lag are not significantly different between two subsets of catchments.

Furthermore, twenty-four climate variables in Table 2 have been used to explore the difference between two subsets of catchments and the possible associations between the magnitude of the shift in CWSC with the values of climate variables. As illustrated in section 3.4, one climate variable consists of four values, i.e., the climate values during the periods before and after the change point, the absolute difference and % variation between the climate variables between two periods. Thus, there were 96 climate values considered in this part. As shown in Figure 8, significant differences (i.e., % variation is larger than $\pm 10$ percentage) have been found in the majority of the climate variables between two subsets of catchments for both four categories of climate values, except for the % difference in the daily maximum temperature, which was not significant for both four values between two periods. In addition, the % difference of the drought length ($\geq 7$ years) between the two subsets of catchments was 14.3%, and on average, the subset of catchments with significantly



increased shift had longer drought length than the other subset.
The potential associations between the change in the CWSC and both climate values have been
presented in Figure 9. A medium degree of correlation (PCC>0.4) has been found between the %
shift in CWSC with the CV of annual rainfall (PCC=0.422), the CV of annual runoff (PCC=0.419)
during the period before the change point. No more than a medium degree correlation has been
found between the shift in CWSC with the climate values after the change point. As for the variation
of climate value between two periods, a larger association has been found in the shift in CSWC
with the variation in daily rainfall (PCC=-0.425), variation in annual rainfall (PCC=-0.518), and
change in annual runoff ratio (PCC=0.479) rather than others. In addition, it seems that the shift in
parameter  $\theta_1$  was not related to drought length (≥7 years) because its PCC value was only 0.148.
## 4.5. Factors for the time-lag of catchment
Using the same method as illustrated in section 4.4, the difference between two subsets of
catchments, and potential associations between the time-lag of the catchment (time-lag) with
catchment properties and climate characteristics were analyzed. In other words, are catchments
with certain catchment properties and/or climate conditions easier to have longer time-lag? It
should be noted that the catchments with a non-significant change in the CWSC were not included
in this part, because these catchments did not experience a significant change in its estimation
value of  $\theta_1$  thus did not have a statistically significant change point.
On average, the time-lag in the subset of catchments with the significant upward change in
estimated  $\theta_1$  was 9.4% larger than the subset with a significant downward shift. As shown in
Figure 10, only lower associations have been found between the time-lag with different catchment.
The maximum PCC value between the time-lag with the catchment properties was just 0.159,
which was achieved by the correlation between the time-lag and the AWHC in top soil. In addition,
the potential association between the time-lag with the climate variables also have been presented
in Figure 11. Similarly, low correlation (|PCC < 0.3|) has been found between the time-lag and
both four categories of climate values.
# 5. Discussions
The results indicate that, under certain circumstances, a long-term meteorological drought would
result in a significant change in the CWSC. In this study, 52 in 83 catchments (62.7%) have been
found to have a significant shift in its CWSC. Furthermore, a subset of 40 catchments had a
significant upward change in CWSC while another subset of 12 catchments had a significant
downward change.
## 5.1. Possible reasons for different changes in the CWSC
Our results indicate that the geographical distribution of two groups of catchments with significant
and non-significant changes in  $\theta_1$  showed a little wide clustering phenomenon while the
geographical distribution of two subsets of catchments with significant upward and downward
changes showed some tendency for clustering. The adjacent catchments tend to have similar


change directions, such as the catchments in southwestern and southern Victoria that both
experienced an upward change in the CWSC while the majority of catchments in northeastern
Victoria experienced a downward change in the CWSC. However, the magnitude of the shift varies
with the individuals. Furthermore, significant differences have been found between two groups of
catchments (catchments with significant and non-significant changes) and two subsets of
catchments (catchments with significant upward and downward changes) in terms of the majority
of catchment properties and climate characteristics. However, no strong association has been
found between the magnitude of the change in the CWSC with these single variables. In addition,
the length of the dry period was not connected to the shift in the CWSC. Thus, it seems that it is
the combination of local catchment properties and climate characteristics rather than a single
factor, controls the catchment response behavior to long-term meteorological drought.
5.1.1 Potential mechanisms for the impacts of catchments properties on the CWSC
The CWSC of a catchment is the comprehensive presentation of catchment properties in the field
of water storage. The interrelated impacts by the changes in catchment properties, such as
groundwater decline, may result in the potential shift in CWSC. However, it should be mentioned
that even similar changes take part in catchments with different backgrounds of properties, may
have opposite impacts on its change directions of CWSC.
In the study area, the groundwater decline combined with different background of soil types in the
catchments may be one of the possible reasons for different change directions in the CWSC. The
groundwater decline would lead to the loss of the hydraulic connection between groundwater and
surface water. The space that once was occupied by soil water become void. However, catchments
with different soils would have different change directions. In sand and other soil types with lower
adhesive property, these soil pores would be compacted due to the reduction of buoyancy of soil
water, thus the compacted soil may result in a decrease in CWSC. Conversely, these soil pores may
be retained in those soils with strong adhesive property, the decline of groundwater may lead to
an increase in CWSC. A significant decline in groundwater level has been observed in southeastern
Australia during the drought periods (Leblanc et al., 2009). Figure 12 presents the soil types of 83
catchments in southeastern Australia, and illustrates that the silt loam and loam are the main soil
types in the study area, the sum of which occupies more than 80 percent of the region. Moreover,
southwestern Victoria and southern New South Wales (i.e., catchments with an increase in the
CWSC), were mainly occupied by the loam while eastern Victoria (i.e., catchments with an increase
in the CWSC) was mainly occupied by the silt loam. By contrast, the silt loam had a stronger
adhesive property and larger field capacity than the loam. Thus, the distribution of these two soil
types may explain a proportion of variance in the change direction of CWSC. However, in southern
Victoria, the results disagree with this finding, the possible reasons might be that there were other
more influential factors that control the catchment behaviors in this region.
In addition, another possible reason is the variance in land use. The primary land-use types in the
study region are Evergreen Broadleaf Forest (49.8%), Wooded Grassland (16.9%), Woodland
(14.1%) and cropland (13.2%). As shown in Figure 13, catchments with downward changes are
mainly covered by the Evergreen Broadleaf Forest while those with upward changes are mainly
covered by other three land-use types. Historical literature (Adams et al., 2012;Fensham et al.,
2009;Ferraz et al., 2009) showed that, due to the persistent drought, plant mortality and change





in species compositions have been observed in southeastern Australia. Thus, it can be
hypothesized that the Evergreen Broadleaf Forest has less resilience in response to the drought
and may be much easier to experience significant changes in the CWSC than other types since that
the growth of the Evergreen Broadleaf Forest needs much more water than other types. In
catchments with high coverage of the Evergreen Broadleaf Forest, the canopy interception and
absorption of the forest usually consist of the vital proportions for the catchments to storage water,
thus the tree die-off in these catchments might result in a decrease in the CWSC. On the contrary,
in the catchment with other land use types, the water storage by its vegetation may only play a
non-primary role and its vegetation has stronger resilience in response to the drought because of
less water consumption. Thus, the persistent drought in these catchments did not result in massive
tree morality but merely led to the increased water stress and the augment of the CWSC.
5.1.2 Potential mechanisms for the impacts of climate characteristics on the CWSC
The effect mechanism of climate characteristics on the CWSC is through their remodeling on the
catchment properties. A medium degree of correlation has been found between the % shift in the
CWSC with the coefficients of annual rainfall and runoff before the change point, and the variation
in annual or daily rainfall and annual runoff ratio. Furthermore, the shift in the CWSC was not
associated with all climate variables after the change point. It also has been found that the annual
distribution of baseflow, the interannual, seasonal and monthly distribution of rainfall and runoff,
are not correlated to the shift in CWSC. Previous studies (Chiew et al., 2011;Saft et al., 2015)
illustrated that during the meteorological drought large reduction in autumn rainfall and an even
larger decrease in winter runoff and annual runoff of many catchments in southeastern Australia
have been observed. However, in our study, significant declines in rainfall and runoff of both four
seasons have been observed throughout the study area, not merely the autumn rainfall and winter
runoff. Furthermore, significant differences have been found in the most climate variables between
catchments with significant upward and downward changes, including the autumn rainfall, and
winter and annual runoff and other climate variables. Thus, it is really hard to judge the influence
of each factor on the CWSC. According to our study, it seems that the final changes in the CWSC
are the combined effects of multiple climate variables and catchment properties.
## 5.2. Catchments with quick or slow response
The length of time-lag represents the resilience of a catchment in response to a prolonged drought.
Our results indicated a negative association between the length of time-lag with the forest
coverage percentage in both catchments with significant upward and downward changes. It means
that catchments with larger forest coverage percentage are more susceptible to the stress from a
chronic meteorological drought than other catchments. This phenomenon is possibly related to
the primary land use types in the study area, i.e., the Evergreen Broadleaf Forest (49.8%), which
has a high demand for water consumption. Thus, a catchment that experienced a prolonged
meteorological drought, combined with the characteristic of high coverage of the Evergreen
Broadleaf Forest, would be quite sensitive to have changes in its CWSC. In addition, opposite
directions of PCC association between the time-lag with several catchment properties (i.e., mean
elevation and elevation difference, mean slope and slope range, AWHC in sub soil, Ks in top soil)
have been found in two subsets of catchments. For example, in the subset of catchments with a





significant decrease in the CWSC, the length of time-lag is positively associated with the elevation
level and slope, while in the subset of catchments with a significant increase in the CWSC, it is a
negative association. Similar findings also have been found in associations between the time-lag
and multiple climate variables (e.g., daily rainfall, baseflow and so on). However, since only a low
association level has been observed between the time-lag and these single climate variables, it is
still hard to judge whether there are certain physical mechanisms behind this phenomenon or it is
just a statistical artifact.

## 6. Conclusions

In this study, changes in the estimated value of a model parameter $\theta_1$ were used to represent the
possible fluctuation in the catchment water storage capacity (CWSC) as a result of a persistent
meteorological drought. $\theta_1$ denotes the CWSC in the model structure. The possible changes in
the CWSC, as well as the time-lag of the catchment (i.e., the time-lag between the onset of the
meteorological drought with the change point of the CWSC), have been studied based on the
catchments in southeastern Australia where a long-term meteorological drought was throughout
the area. Significant changes in CWSC have been identified in 62.7% (52 in 83) of catchments.
Furthermore, 48.2% (40) of catchments experienced a significant increase in CWSC while 14.5%
(12) of catchments experienced a significant decrease in the CWSC during the drought period.
Different change directions in the CWSC resulted in the opposite impacts on runoff generation, i.e.,
catchments with increased CWSC would result in lower runoff generation rates for similar amounts
of rainfall than before while those catchments with decreased CWSC would have an opposite
response (higher runoff generation rate). It seems that the final changes in the CWSC are the
combined effects of multiple climate variables and catchment properties. Generally, catchments
with larger elevation and slope, lower forest coverage (Evergreen Broadleaf Forest) are more likely
to have an upward change in the CWSC within a chronic drought while smaller catchments with
higher elevation, lower coverage of the Evergreen Broadleaf Forest are more likely to have a
downward change in the CWSC. Among all catchment properties and climate variables considered,
our results suggest that two climate variables (i.e., variation in annual rainfall and annual runoff
ratio) have the strongest associations with the shift in the CWSC. In addition, the changed
catchments were not equally susceptible to the pressure due to persistent meteorological drought.
Catchments with a lower proportion of Evergreen Broadleaf Forest usually have longer time-lag
and are more resilient. This study gives us a better understanding of possible changes in catchment
water storage capacity induced by a prolonged meteorological drought, which will help improve
our ability to simulate the hydrological system. However, several unsolved issues are stilled
remained that need to be addressed in future research. Above all, the changes in catchment water
storage capacity are the combined effects of multiple catchment properties and climate
characteristics, further studies are still required to confirm which factors played the most
important role in the catchment dynamic. Secondly, this study used one conceptual rainfall-runoff
model and its parameter to model and represent the catchment water storage capacity, our
findings should be verified with other methods. Third, this paper presents a method to simulate
the possible changes in catchment water storage capacity, further research is needed to explore
whether it is possible to predict the potential changes in catchment water storage capacity in the
future.



## Acknowledgments

This research is funded in part by the National Key Research and Development Program (2018YFC0407202), the National Natural Science Foundation of China (51861125102; 51879193; 41890822), and the Natural Science Foundation of Hubei Province (2017CFA015). The authors appreciate the help from the Supercomputing Center of Wuhan University for providing necessary guides to perform the numerical calculations of this study on the supercomputing system.

## Author contributions

Zhengke Pan and Pan Liu conceived the study and wrote the paper. Chongyu Xu, Lei Cheng, and Jing Tian made constructive comments on the writing of this study, which improves the quality of this paper. Shujie Cheng and Kang Xie provided the data of the catchment attributes and made comments. All of the authors read and approved the manuscript.

## Code/Data availability

The data and codes that support the findings of this study are available from the corresponding author upon reasonable request.

## Compliance with ethical standards

**Conflict of interest:** The authors declare that they have no conflict of interest.

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



## Tables

Table 1. Prior ranges for all unknown parameters in the GR4J hydrological model

| Parameters | Min | Max |
|:---:|:---:|:---:|
| $\theta_1$ | 1.0 | 500.0 |
| $\theta_2$ | -5.0 | 5.0 |
| $\theta_3$ | 1.0 | 200.0 |
| $\theta_4$ | 0.1 | 8.0 |




Table 2. Potential factors for exploring the association between the change of in CWSC
with catchment properties and climate characteristics.

| Category | Variables |
|---|---|
| **Catchment properties** | X1. Catchment area (km2) |
| | X2. Elevation difference between the maximum and minimum elevations (m) |
| | X3. Mean elevation (m) |
| | X4. Slope range (deg) |
| | X5. Mean slope (deg) |
| | X6. Forest coverage percentage (%) |
| | X7. AWHC top soil (mm) |
| | X8. AWHC sub soil (mm) |
| | X9 Ks in top soil (mm/h) |
| **Climate characteristics** | Y1. Average of daily rainfall |
| | Y2. Average of daily potential evapotranspiration |
| | Y3. Average maximum daily temperature |
| | Y4. Average minimum daily temperature |
| | Y5. Cv of month rainfall |
| | Y6. Cv of month runoff |
| | Y7. Average of month runoff ratio |
| | Y8. Average of spring rainfall |
| | Y9. Average of summer rainfall |
| | Y10. Average of autumn rainfall |
| | Y11. Average of winter rainfall |
| | Y12. Average of spring runoff |
| | Y13. Average of summer runoff |
| | Y14. Average of autumn runoff |
| | Y15. Average of winter runoff |
| | Y16. Average of annual rainfall |
| | Y17. Average of annual potential evapotranspiration |
| | Y18. Average of annual runoff |
| | Y19. Average of annual aridity index (PET/rainfall) |
| | Y20. Average of annual runoff ratio |
| | Y21. Cv of annual rainfall |
| | Y22. Cv of annual runoff |
| | Y23. Average of annual baseflow |
| | Y24. Annual base flow index |

Note: 1. AWSC is the Available soil Water Holding Capacity; 2. Ks is the saturated hydraulic
conductivity; 3. Cv denotes the coefficient of variation.





Table 3. The direction of the shifts in CWSC due to the long-term meteorological
drought for the catchments in southeastern Australia.

| Magnitude | Change direction | Percentage (Number of catchments) |
|---|---|---|
| Significant change | Downward (Smaller CWSC than the previous estimation suggest) | 8.3% (12) |
| | Upward (Larger CWSC than the previous estimation suggest) | 27.6% (40) |
| Non-significant change | Slight increase | 12.4% (18) |
| | Slight decrease | 9.0% (13) |
| Dissatisfy the criteria of the minimum NSE performance and the maximum performance degradation | | 42.8% (62) |
| All (catchments with a sustained meteorological drought) | | 100% (145) |






## Figures

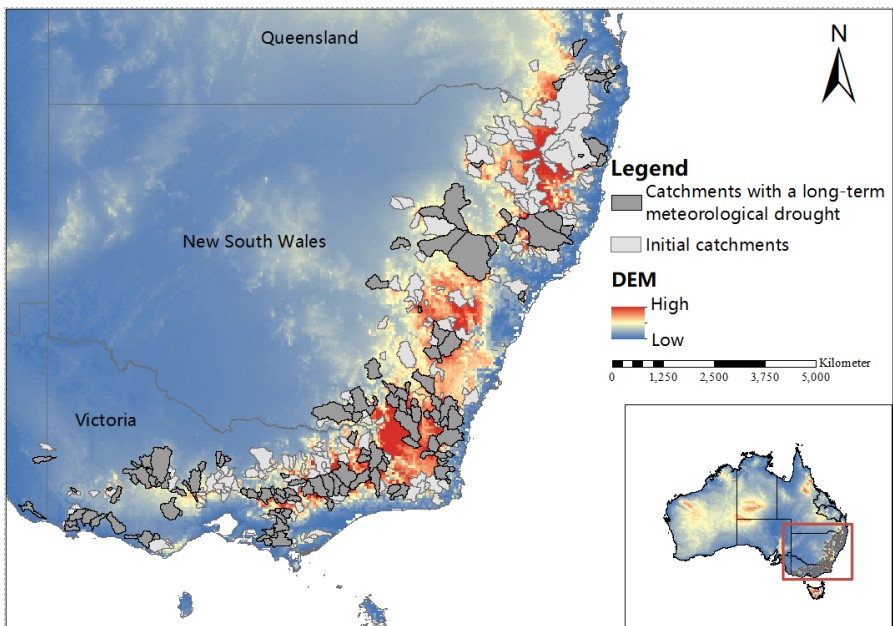

Figure 1. Location map of the study catchments in southeastern Australia. The dark gray color denotes the catchments with a long-term meteorological drought (145 catchments) while the light gray color denotes the catchments without any sustained droughts or has more than one prolonged drought period (253 catchments).

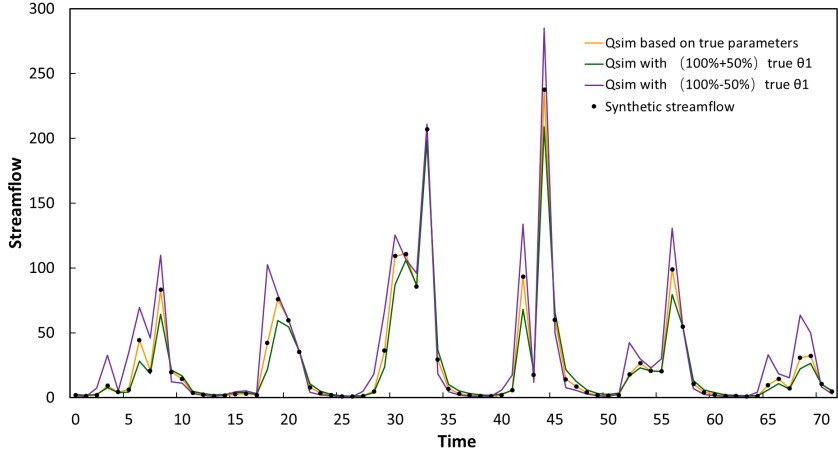

Figure 2. Example of the impacts on model simulation due to the different shifts in model parameter $\theta_1$ (that denotes the catchment water storage capacity)

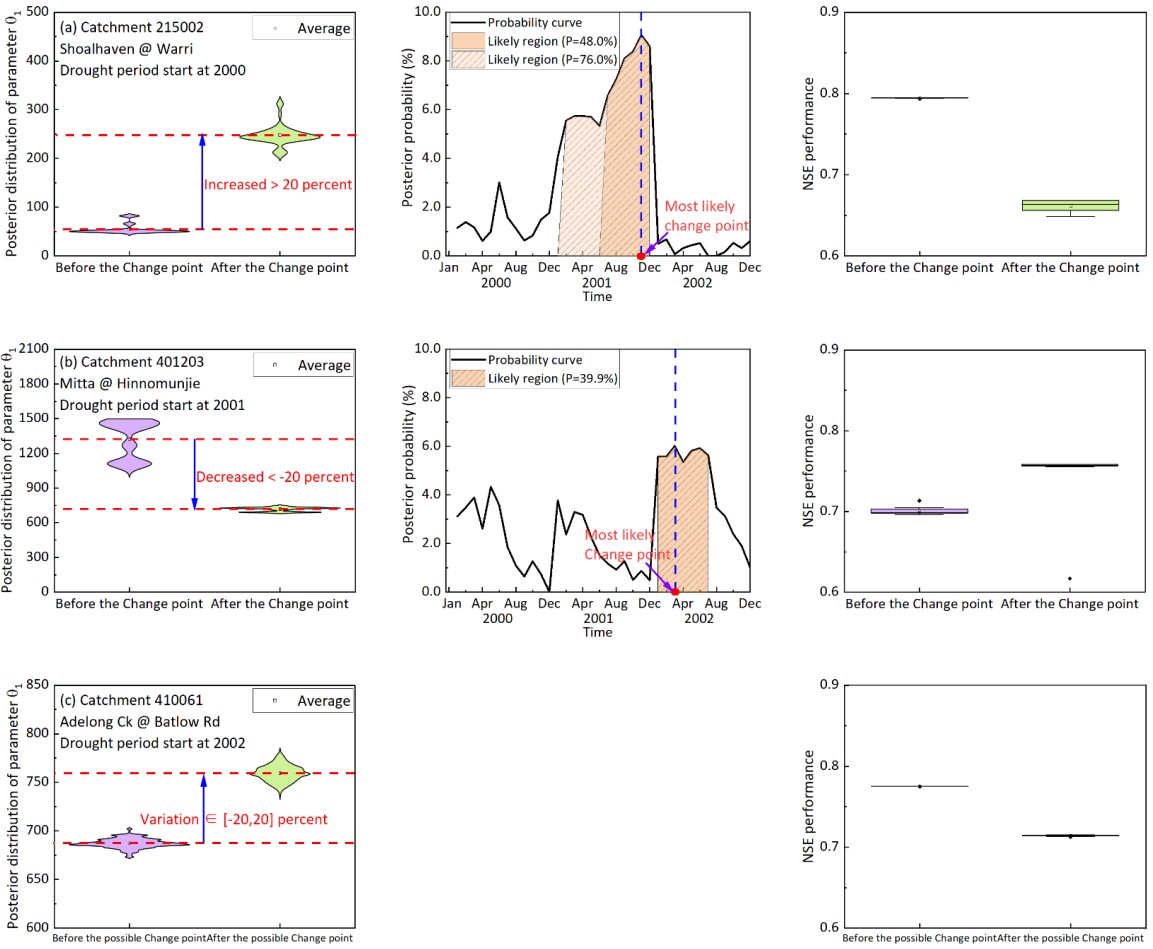

Figure 3. Examples of shifts in model parameter $\theta_1$: (a) catchment 215002 with a significant upward change in $\theta_1$; (b) catchment 401203 with a significant downward change in $\theta_1$; and (c) catchment 410061 with a non-significant change in $\theta_1$. The first column compares the posterior distributions of $\theta_1$ calibrated during the periods before and after the Change point. The second column denotes the posterior probabilities based on all possible Change points. The last column denotes the NSE performances of the model parameters calibrated during the periods before and after the most likely Change point.



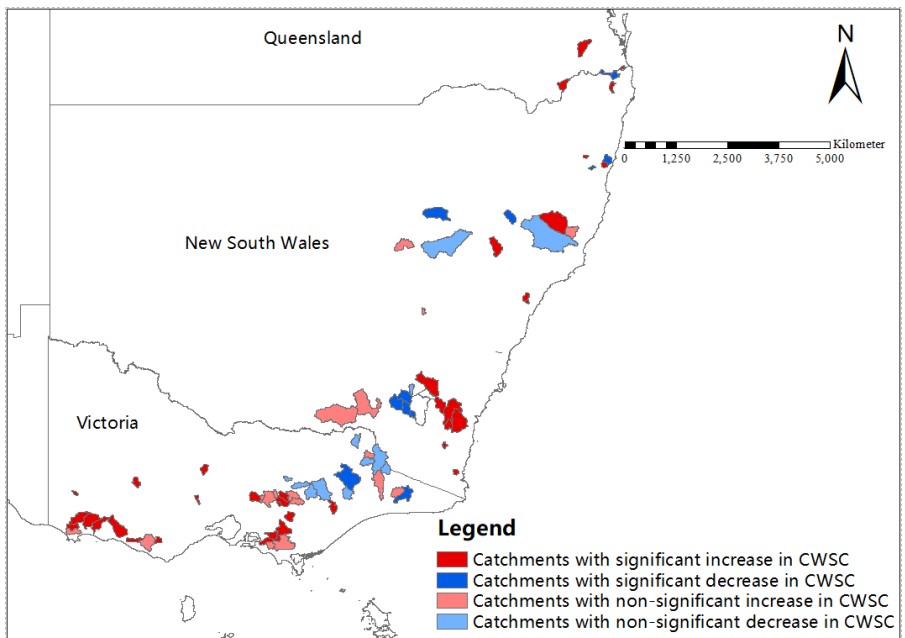

803

Figure 4. Location map of the catchments with the significant and non-significant shifts in the CWSC. The red (pink) color denotes the catchments that have a significant increase (decrease) in the CWSC after the Change point while the (nattier) blue color denotes the catchments with a non-significant increase (decrease) in the CWSC.


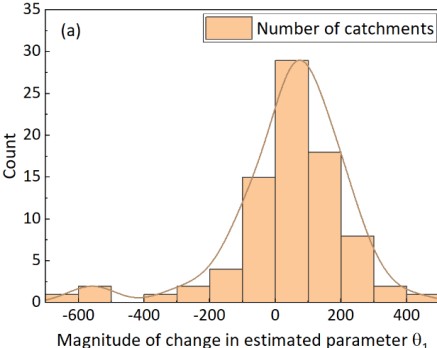
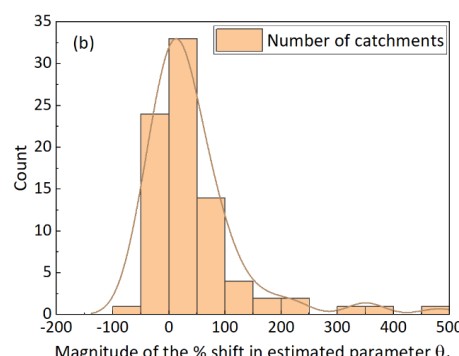

Figure 5. Shift magnitudes of CWSC between the periods before and after the Change
point. The orange lines denote the Kernel Smooth curve of the histograms.


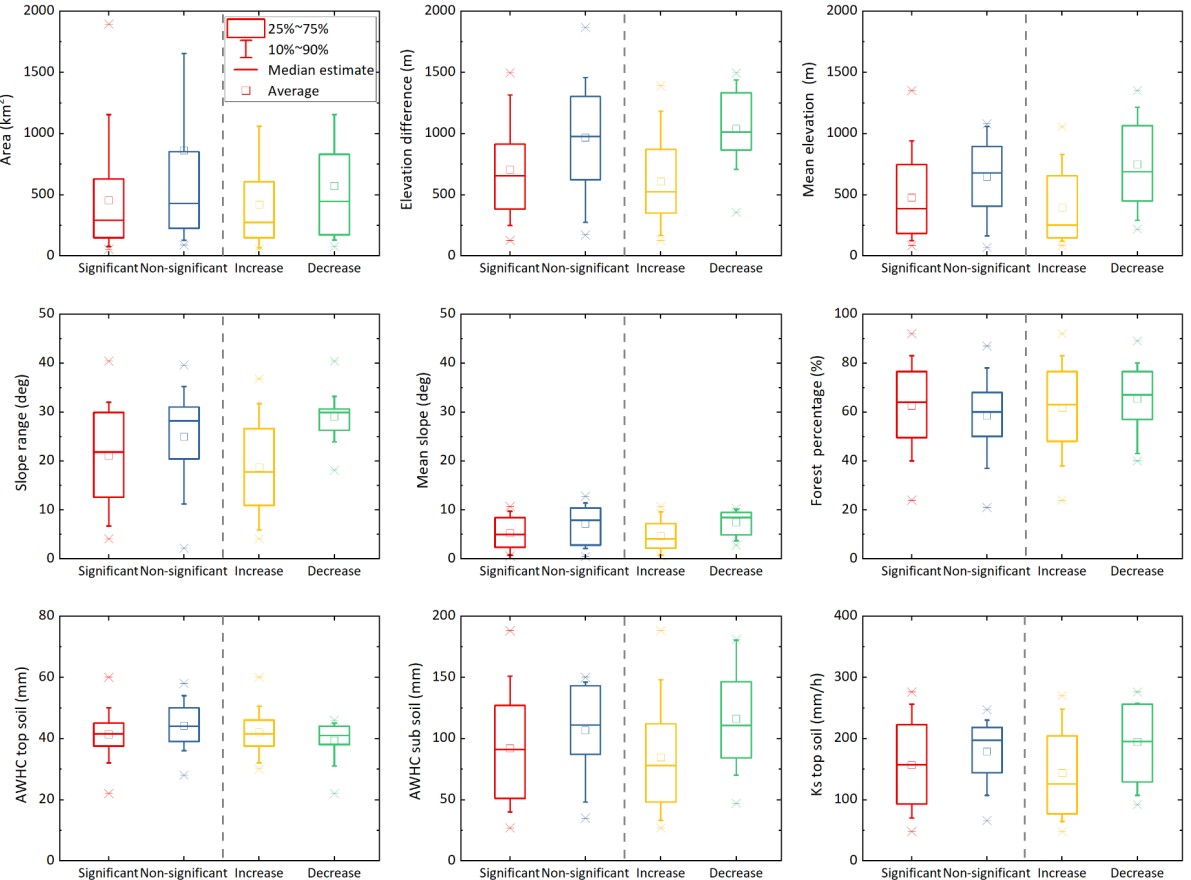

Figure 6. Catchment properties for the study catchments, including catchment area (km$^2$), elevation difference between the maximum and minimum elevations (m), mean elevation (m), slope range (deg), mean slope (deg), forest coverage percentage (%), Available soil Water Holding Capacity (AWHC) in top and sub soils, and saturated hydraulic conductivity (Ks) in top soil (mm/h). The red and black lines (solid) denote the average of catchments with and without significant changes in the CWSC, respectively. The "Increased" means catchments with a significant increase in CWSC while the "Decreased'' represents catchments with a significant decrease in the CWSC.

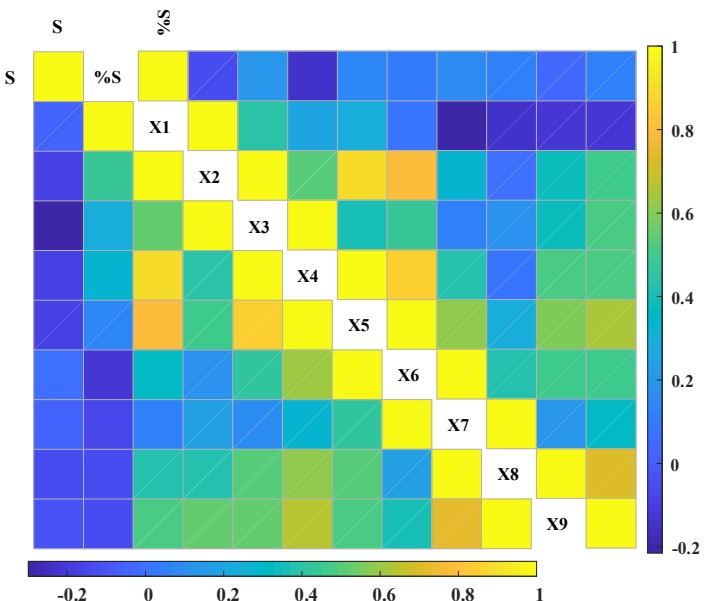

Figure 7. Pearson Correlation Coefficient based on the association between the magnitude of the shift in $\theta_1$ and multiple catchment properties as well as the associations between different catchment properties. In the lower triangular matrix, shift in in $\theta_1$ was considered while in the upper triangular matrix, the percentage shift in $\theta_1$ was used. S denotes the shift in $\theta_1$, %S denotes the percentage shift in $\theta_1$ while X1-X9 denotes the catchment properties mentioned in Table 2.

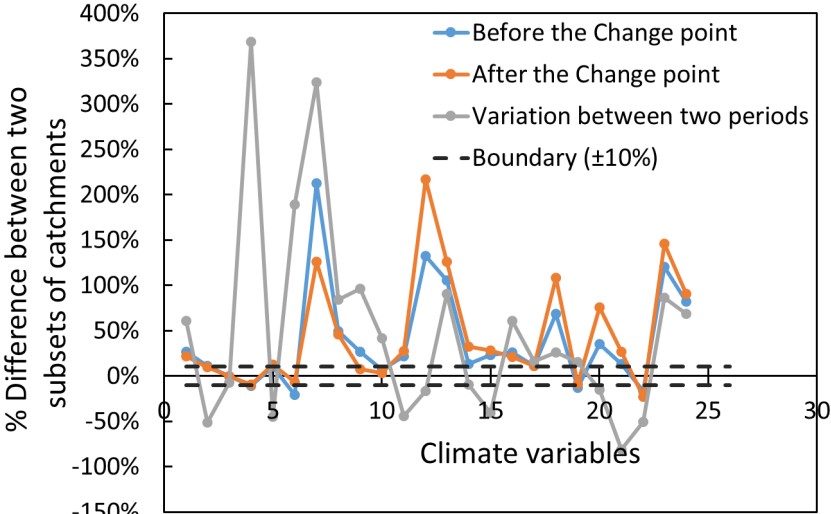

830

Figure 8. The percentage difference of climate variables between two subsets of
catchments (catchments with significant upward and downward changes in estimated
$\theta_1$). The numbers in X-coordinate denote climate variables illustrated in Table 2. The
blue and orange lines denote the percentage difference of climate variables between
two subsets of catchments during the periods before and after the Change point,
respectively. The gray line denotes the percentage difference of the amplitude of
variation between two periods. The positive value means an increase while a negative
value means a decrease.

(a) Before the Change point

(b) After the Change point

(c) Amplitude of variation

(d) Amplitude of the % variation

Figure 9. Pearson Correlation Coefficient based on the association between (1) the
magnitude of shifts in the CWSC with multiple climate variables as well as the
connection between different climate variables (lower triangular matrix) and (2) the
magnitude of % shifts in the CWSC with multiple climate variables as well as the
connection between different climate variables (upper triangular matrix). Sub figures
(a) and (b) denote a connection between the shift or % shift in the CWSC with the
climate variables during the periods before and after the Change point, respectively.
Sub figures (c) and (d) denote the association between the shift or % shift in the CWSC
with the variation and % variation of climate variables of two periods, respectively. Y1-
Y24 denotes the climate variables in Table 2.



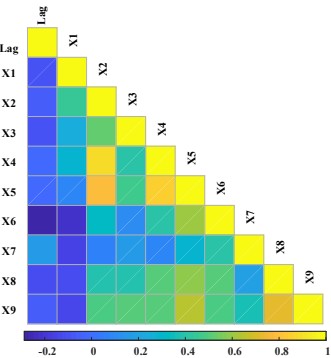

Figure 10. Pearson Correlation Coefficient based on the length of the time-lag of the catchment (between the start of the meteorological drought and the Change point) with the catchment properties.

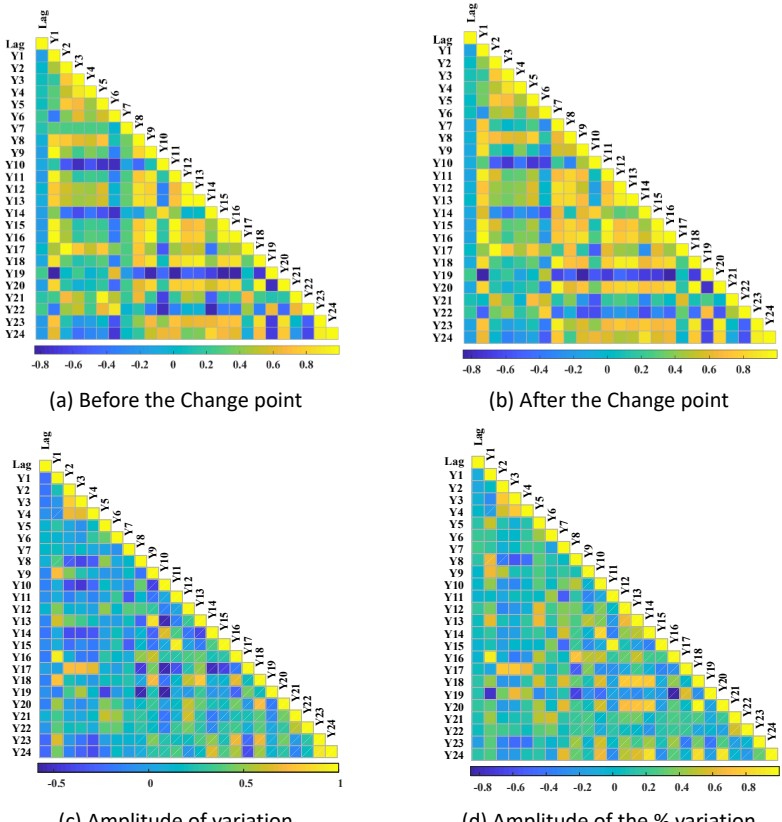

Figure 11. Pearson Correlation Coefficient based on the association between the length of the time-lag of the catchment with multiple climate variables.

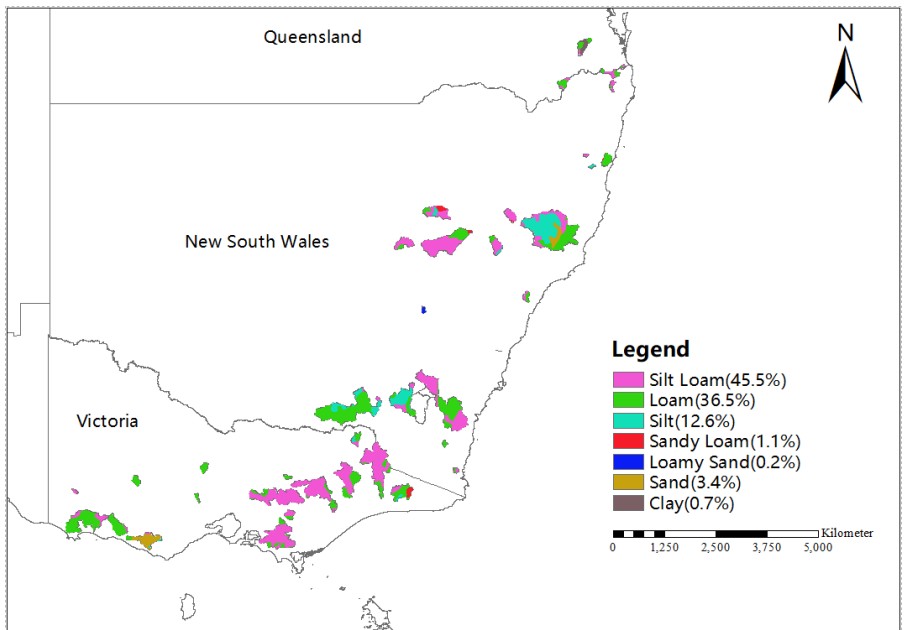

863

Figure 12. Soil types and corresponding percentages in the study catchments (83
catchments). The data of soil types was adopted from the Harmonized World Soil
Database                (http://www.fao.org/soils-portal/soil-survey/soil-maps-and-
databases/harmonized-world-soil-database-v12/zh/, Fischer et al. (2008)) and
classified according to the Soil Texture Triangle of USDA.

869



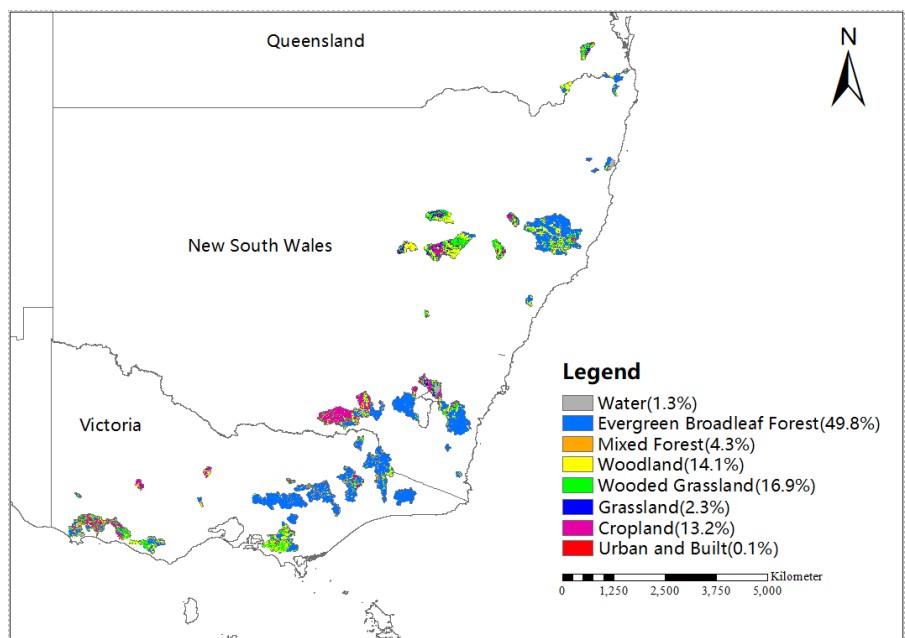

Figure 13. Land use types and corresponding percentages in the study catchments (83 catchments). The data of land use was adopted from the global land cover map at 1 km resolution released by the University of Maryland (http://app.earth-observer.org/data/basemaps/images/global/LandCover_512/LandCoverUMD_512/LandCoverUMD_512.html?tdsourcetag=s_pctim_aiomsg, Hansen et al. (2000))