# Peer review of "The Influence of a Prolonged Meteorological Drought on the Catchment Water Storage Capacity: A Hydrological Model Perspective"

_Hydrology and Earth System Sciences, 2019_

## Referee Comment (RC1) · Anonymous Referee #1 · 17 Mar 2020

This paper represents an interesting huge work, the main objective of this paper is to detect the possible changes in catchment water storage capacity induced by a prolonged meteorological drought. It will also help improve our ability to simulate the hydrological system. Meanwhile, the structure is clear and analyses and conclusions in this paper are reasonable and logical. Overall, it is a nice paper with proper methodology and complete discussion on the main findings. A minor modification should be considered before publication.

General comments: First, the results indicated that 12 catchments had a significant
downward shift in the CWSC. My first principal doubts come from there: are 12 sufficient to be able to analyze regularity about catchments with a significant downward shift. Second: Formulating conclusions is similar to drawing up an abstract. I mean that the emphases of the abstract and conclusions are different.

Detailed comments: * Is there a mistake in the inconsistency between the Abstract (L37-L38) and Conclusion (L613-615). * Previous study (Yan et al., 2015) has indicated that the sensitivity of the parameters in the GR4J model has different performance in different parts of the same basin. Therefore, you should cite more citations or do some researches in the study area to support the sentence in the L224-L226. * L503-517: You should cut out much of the repetition with the Section 4. * In the Section 4.4ïïjŽYou had better discuss the factors for the direction first and then the magnitude of shifts in the CWSC. * Why not analyze the reasons for conclusion 2 in the Section 5.1? (Catchments with larger elevation and slope, lower forest coverage of Evergreen Broad leaf Forest are more likely to have an increase in the CWSC during a chronic drought while smaller catchments with lower elevation, lower coverage of the Evergreen Broad leaf Forest are more likely to have a decrease in the CWSC.) * The conclusion should be concise and coherent. It is suggested that only the key and definitive conclusions of this paper should be stated. If you cannot confirm your conclusion, these results may only be used in the Discussion, such as L610-611 and L622-631, etc. * It's better to add latitude and longitude to the location map of the study catchments (Figure 1). * Table 3 should add the results about the fourth criteria. * Figure 4 can be improved. The figure should show the change directions and magnitudes of the CWSC at the same time. It is better for readers to understand the shifts of the CWSC in the adjacent catchments. * Is the supplementary material missing?

Reference Yan, X. et al., 2015. Application of GR4J Rainfall-runoff Model to Typical Catchments in the Yellow River Basin. Proceedings of the 5th International Yellow River Forum on Ensuring Water Right of the River's Demand and Healthy River Basin Maintenance, Vol V, 191-198 pp.

---

## Referee Comment (RC2) · Anonymous Referee #2 · 29 Apr 2020

I read with interest the paper on The Influence of a Prolonged Meteorological Drought on the Catchment 1 Water Storage Capacity: A Hydrological Model Perspective. It's a good piece of work whose findings are useful for water resources management in the light of climate and landuse changes.

My major concern however is in the study design and methodology. Most of the steps in the methods section are not adequately described. Readers can fully link the methods with the results because certain information is not provided in the methods section. Typical examples but not limited to: Which datasets have been used in all the objectives

Describe the spatial and temporal variation of these datasets What are the different sources of datasets For examples what is the source of the primary land-use types described in the results section.

Authors mention about the catchment physical properties and climate characteristics which influences CWSC. Readers will only get to understand or know these physical properties and climate characteristics when they read the results section. This again has to do with a poor study design.

In section 3.4, what do authors mean by '…..because of the limitation of available data of catchment attributes, only unique catchment properties are employed…". What are these unique catchment properties? And how were they selected?

The authors use the term 'climate characteristics' in the abstract and methods section- Upon reading the manuscript (rather the results) one discovers that its only rainfall being referred to. Is rainfall enough to define meteorological characteristics? - what about the influence of other climate characteristics such as Temperature, Evapotranspiration?

In section 4.3 authors point out that 'However, the geographical distribution of catchments with significant and non-significant changes in ðİIJČ1 showed no obvious geolocation clustering phenomenon.' Which statistical techniques did the authors employ to come up with conclusions on significance in ðİIJČ1 Given multiple climate variables and catchment properties-which one significantly affects CWSC?

---

## Author Comment (AC1) · 27 Jun 2020

Responses to Referee #1: This paper represents an interesting huge work, the main objective of this paper is to detect the possible changes in catchment water storage capacity induced by a prolonged meteorological drought. It will also help improve our ability to simulate the hydrological system. Meanwhile, the structure is clear and analyses and conclusions in this paper are reasonable and logical. Overall, it is a nice paper with proper methodology and complete discussion on the main findings. A minor modification should be considered before publication.

Reply: We are grateful for the reviewer's appreciation of our work and for the professional comments, which are carefully followed in making revision.

General comments: A1: First, the results indicated that 12 catchments had a significant downward shift in the CWSC. My first principal doubts come from there: are 12 sufficient to be able to analyze regularity about catchments with a significant downward shift. Reply: Thanks. We will make it clearer in the revised version that to ensure the accuracy of the modeling results, strict criteria were adopted to evaluate the model results and to identify the catchments with a significant change in $\theta\_1$ including the minimum NSE requirement, the minimum requirement of significant change, the requirement for maximum performance degradation, and the requirement for the robustness of results (see section 3.2.5). Only these catchments that satisfied all these criteria were identified as catchments undergo significant changes. Actually, only 12 of the 145 catchments, which had experienced a long-term meteorological drought, satisfied all these four criteria and have been identified with a significant downward shift in the CWSC. Thus, the identified 12 catchments that had a significant downward shift should be reasonable results. We will also clarify that this result depends on the criteria used for evaluation. All catchments from southeastern Australia in the dataset have been tested in this study using the above defined criteria, and we did not find more catchments with a significant downward shift in the CWSC. More explanation will be provided in the revision.

A2: Second: Formulating conclusions is similar to drawing up an abstract. I mean that the emphases of the abstract and conclusions are different. Reply: Thank you for your comments, following which, the Abstract and conclusion will be rewritten. The part of Abstract will be modified as follows: Understanding the propagation of prolonged meteorological droughts helps solve the problem of intensified water scarcity around the world. Most of the existing literature studied the propagation of drought from one type to another (e.g., from meteorological to hydrological drought) with statistical approaches, there remains a difficulty in revealing the causality between the meteorological drought

and potential changes in the Catchment Water Storage Capacity (CWSC). This study aims to identify the response of the CWSC to the meteorological drought by examining the changes of hydrological model parameters after the drought events. Firstly, the temporal variation of a model parameter that denotes the CWSC is estimated to reflect the potential changes in real CWSC. Next, the change points of the CWSC parameter were determined based on the Bayesian change point analysis. Finally, the possible association and linkage between the shift in the CWSC and the time-lag of the catchment (i.e., time-lag between the onset of the drought and the change point) with multiple catchment properties and climate characteristics were identified. In total 145 catchments from southeastern Australia were selected as the study areas. Results indicated that (1) significant shifts in the CWSC can be observed in 62.7% of the catchments, which can be divided into two subsets with the opposite response, i.e. 48.2% of catchments had lower runoff generation rates while 14.5% of catchments had higher runoff generation rate; (2) the increase in the CWSC during a chronic drought can be observed in smaller catchments with lower elevation, slope, and forest coverage of Evergreen Broadleaf Forest, while the decrease in the CWSC can be observed in larger catchments with higher elevation and larger coverage of the Evergreen Broadleaf Forest; (3) catchments with a lower proportion of Evergreen Broadleaf Forest usually have longer time-lag and are more resilient. This study improves our understanding of possible changes in the CWSC induced by a prolonged meteorological drought, which will help improve our ability to simulate the hydrological system under climate change. The Conclusion will be modified as follows: This study aims to examine the possible changes in the CWSC as well as the time-lag between the onset of the meteorological drought and the change point of the CWSC. A classical hydrological model, GR4J, was used and its parameter $\theta\_1$ was selected to denote catchment water storage capacity (CWSC). Thus, the temporal variation in parameter $\theta\_1$ was detected to reveal the possible fluctuation in the CWSC, and the causality between the temporal variation in parameter $\theta\_1$ and a persistent meteorological drought was examined. One hundred forty-five catchments in southeastern Australia were selected as the study areas. Main conclusions can be drawn as follows: (1) Significant changes in the CWSC have been identified in 62.7% (52 in 83) of catchments, which can be divided into two subsets with opposite catchment responses: 48.2% (40 in 83) experienced a significant decrease in the CWSC during the drought period and had lower runoff generation rates, while 14.5% (12 in 83) of catchments experienced a significant decrease in the CWSC during the drought period, and had higher runoff generation rate. (2) Different change directions in the CWSC resulted in the opposite impacts on runoff generation, i.e., catchments with increased CWSC would result in lower runoff generation rates for similar amounts of rainfall than before while those catchments with decreased CWSC would have an opposite response (higher runoff generation rate). Generally, the increase in the CWSC during a chronic drought can be observed in smaller catchments with lower elevation, slope, and forest coverage of Evergreen Broadleaf Forest, while the decrease in the CWSC can be observed in larger catchments with higher elevation and larger coverage of the Evergreen Broadleaf Forest. Among all catchment properties and climate variables considered, our results suggest that two climate variables (i.e., variation in annual rainfall and annual runoff ratio) have the strongest associations with the shift in the CWSC. (3) The responses of different catchments to persistent meteorological drought were not equally susceptible. Catchments with a lower proportion of Evergreen Broadleaf Forest usually have longer time-lag and are more resilient. It is noted that although this study resulted in interesting findings that give new insight and have not been fully outlined before, it is based on the lumped GR4J model and the specific case in Australia, which implies that the main findings/conclusions may not directly extendable to other regions. Thus, to examine the generality of the main conclusions, the response of CWSC to the meteorological drought can be analyzed with the other hydrological models in the other regions.

Detailed comments: A3: Is there a mistake in the inconsistency between the Abstract (L37-L38) and Conclusion (L613-615). Reply: We are sorry for this error. The sentence in the Abstract and Conclusion will be modified as "...the increase in the CWSC during a chronic drought can be observed in smaller catchments with lower elevation, slope,

and forest coverage of Evergreen Broadleaf Forest, while the decrease in the CWSC can be observed in larger catchments with higher elevation and larger coverage of the Evergreen Broadleaf Forest.".

A4: Previous study (Yan et al., 2015) has indicated that the sensitivity of the parameters in the GR4J model has different performance in different parts of the same basin. Therefore, you should cite more citations or do some researches in the study area to support the sentence in the L224-L226. Reply: Thank you. More citations will be added to the revised manuscript. This sentence will be modified as "Previous studies (Demirel et al., 2013; Pan et al., 2019a; Pan et al., 2019b; Perrin et al., 2003; Westra et al., 2014; Yan et al., 2015) showed that $\theta\_1$, which denotes the catchment water storage capacity, is the most sensitive parameter in the structure of the GR4J model.". Added reference: Yan, X. L., Zhang, J. Y., Wang, G. Q., Bao, Z. X., Liu, C. S., and Xuan, Y. Q.: Application of GR4J Rainfall-runoff Model to Typical Catchments in the Yellow River Basin, Proceedings of the 5th International Yellow River Forum on Ensuring Water Right of the River's Demand and Healthy River Basin Maintenance, Vol V, edited by: Hongqi, S., and Xiangxin, L., Yellow River Conservancy Press, Zhengzhou, 191-198 pp., 2015. Demirel, M. C., Booij, M. J., and Hoekstra, A. Y.: Effect of different uncertainty sources on the skill of 10 day ensemble low flow forecasts for two hydrological models, Water Resour. Res., 49, 4035-4053, 10.1002/wrcr.20294, 2013.

A5: L503-517: You should cut out much of the repetition with the Section 4. Reply: Thanks. Changes will be made as suggested. The paragraph in lines 503-517 will be modified as follows: The results indicate that, under certain circumstances, a long-term meteorological drought would result in a significant change in the CWSC. However, no strong association has been found between the magnitude of the change in the CWSC with any single variable. In addition, the length of dry period was not associated with the shift in the CWSC. Thus, it seems that the catchment response behavior to long-term meteorological drought is controlled by the combination of local catchment properties and climate characteristics rather than a single factor. Thus, further studies are still required to confirm which factors played the most important role in the catchment dynamic. A6: In the Section 4.4 You had better discuss the factors for the direction first and then the magnitude of shifts in the CWSC. Reply: Thanks. We will explain better that Section 4.4.1 presents the results that classify the catchments into two groups, one group shows significant shifts in the CWSC, and another does not. While section 4.4.2 continued to discuss the difference between two sub-groups of catchments with significant upward/significant downward shifts in the CWSC. These two sub-groups of catchments were extracted from the group of catchments with significant shifts in the CWSC according to the change direction of the estimated $\theta\_1$. Thus, we thought it might be better to discuss the difference between two groups of catchments with/without significant shifts at first, and then illustrate the difference between two subsets of catchments with significant upward/significant downward shifts later. To make it clearer, after considering reviewer's advice and reminder, the headings of sections 4.4.1 and 4.4.2 will be modified as "4.4.1 Factors for the significant/non-significant shifts in the CWSC" and "4.4.2 Factors for the significant upward/downward shifts in the CWSC" in the revised manuscript.

A7: Why not analyze the reasons for conclusion 2 in the Section 5.1? (Catchments with larger elevation and slope, lower forest coverage of Evergreen Broad leaf Forest are more likely to have an increase in the CWSC during a chronic drought while smaller catchments with lower elevation, lower coverage of the Evergreen Broad leaf Forest are more likely to have a decrease in the CWSC.) Reply: Thanks. We tried but as shown in Figures 7 and 9, no strong PCC association has been found between the magnitude of the change in the CWSC and the single catchment property or climate variable. It seems that the catchment response behavior to long-term meteorological drought is controlled by the combination of local catchment properties and climate characteristics rather than a single factor Thus, we prospected that further studies are still required to confirm which factors played the most important role in the catchment dynamic.

A8: The conclusion should be concise and coherent. It is suggested that only the key

and definitive conclusions of this paper should be stated. If you cannot confirm your conclusion, these results may only be used in the Discussion, such as L610-611 and L622-631, etc. Reply: Thanks. Changes will be made as suggested. The sentences in lines 610-611 will be deleted in the conclusion part of the revised manuscript. In addition, the sentences in lines 622-631 will be modified as follows: It is noted that although this study resulted in interesting findings that give new insight and have not been fully outlined before, it is based on the lumped GR4J model and the specific case in Australia, which implies that the main findings/conclusions may not directly extendable to other regions. Thus, to examine the generality of the main conclusions, the response of CWSC to the meteorological drought can be analyzed with the other hydrological models in the other regions.

A9: It's better to add latitude and longitude to the location map of the study catchments (Figure 1). Reply: Thanks. Changes will be made in the revised Figure 1.

A10: Table 3 should add the results about the fourth criteria. Reply: Thanks. Changes will be made in the revised Table 3. The modified Table 3 is presented in the Supplement.pdf.

A11: Figure 4 can be improved. The figure should show the change directions and magnitudes of the CWSC at the same time. It is better for readers to understand the shifts of the CWSC in the adjacent catchments. Reply: Thanks. Changes will be made as suggested in The modified Figure 4 ïïjĹFigure 2 inïïjĽ.

A12: Is the supplementary material missing? Reply: We are sorry for this mistake. We have uploaded the supplementary material (Table S1) to the system. Please also refer to the supplement.pdf (Table S1).

Please also note the supplement to this comment:
https://www.hydrol-earth-syst-sci-discuss.net/hess-2019-692/hess-2019-692-AC1-supplement.zip

[Figure]

[Figure]

Fig. 1. Figure 1. Location map of the study catchments in southeastern Australia.

[Figure]

Fig. 2. Figure 4. Location map of the catchments with the significant and non-significant shifts in the CWSC.

---

## Author Comment (AC2) · 27 Jun 2020

Responses to Referee #2:

I read with interest the paper on The Influence of a Prolonged Meteorological Drought on the Catchment 1 Water Storage Capacity: A Hydrological Model Perspective. It's a good piece of work whose findings are useful for water resources management in the light of climate and landuse changes. My major concern however is in the study design and methodology. Most of the steps in the methods section are not adequately

described. Readers can fully link the methods with the results because certain information is not provided in the methods section.

Reply: Thanks for the comment and suggestions, we will carefully improve the description of the study design and methodology part in the revision. Some examples are described below: B1: Typical examples but not limited to: Which datasets have been used in all the objectives Describe the spatial and temporal variation of these datasets What are the different sources of datasets For examples what is the source of the primary land-use types described in the results section.

Reply: Thanks. More descriptions of the adopted data set will be added in the revised manuscript. The sentences in lines 159-166 in the original manuscript for describing the research data will be modified as follows: The following data have been used in this study: (1) climate variables, include daily rainfall and daily potential evapotranspiration; (2) daily streamflow observation at catchment outlet; (3) land use types at 1 km resolution; (4) soil types at 30 arc-second resolution, and (5) catchment attributes, include catchment area, mean elevation and so on. The detailed lists of the catchment attributes and climate characteristics were presented in Table 2. The data of climate variables, daily runoff, and catchment attributes were obtained from the Australian Water Resources Assessment (AWRA) system, which has been served as a standard publicly available national dataset for hydrological model evaluation (https://publications.csiro.au/rpr/pub?pid=csiro:EP113194, Zhang et al. (2013)). For all catchments, there is no missing data in the rainfall and potential evapotranspiration data while the runoff data in some catchments are missing. The data set of soil types was obtained from the Harmonized World Soil Database by the Food and Agriculture Organization of the United Nations (http://www.fao.org/soils-portal/soil-survey/soil-maps-and-databases/harmonized-world-soil-database-v12/en/, Fischer et al. (2008)) and was classified according to the Soil Texture Triangle of USDA (https://www.nrcs.usda.gov/wps/portal/nrcs/detail/soils/survey/?cid=nrcs142p2_054167). The data set of the land use types was derived from the global land cover map released by the University of Maryland (UMD) (Hansen et al., 2000) and was classified according to the UMD Land Cover Classification method (http://app.earth-observer.org/data/basemaps/images/global/LandCover_512/LandCoverUMD_512/LandCoverUMD_512.html).

B2: Authors mention about the catchment physical properties and climate characteristics which influences CWSC. Readers will only get to understand or know these physical properties and climate characteristics when they read the results section. This again has to do with a poor study design.

Reply: Thanks. The list of catchment properties and climate characteristics that might influence CWSC was presented in Table 2. And the illustrations of these potential factors were presented in section 3.4. To facilitate the understanding of this manuscript, the following sentence will be added in section 2.2. of the modified manuscript: The detailed lists of the catchment attributes and climate characteristics were presented in Table 2.

B3: In section 3.4, what do authors mean by '. . ...because of the limitation of available data of catchment attributes, only unique catchment properties are employed. . .". What are these unique catchment properties? And how were they selected?

Reply: Thanks. (1) The 'unique catchment properties' we meant these data are static values rather than a time series. For example, only one value of the catchment area was used. (2) The unique catchment properties refer to the X1-X9 in Table 2, i.e., catchment area, mean elevation and elevation difference between the maximum and minimum elevations, mean slope, and slope range, forest coverage percentage, saturated hydraulic conductivity of the top soil, available soil water holding capacity of the top soil and sub soil. (3) These catchment attributes were selected because they reflect the physical characteristics of the catchment, and might be related to the shift in the CWSC. (4) To make it clearer, this sentence will be modified as "It should be noted that due to the limitation of available data of catchment attributes, for each catchment, only one static/constant value of the catchment property was employed (X1-X9).".

B4: The authors use the term 'climate characteristics' in the abstract and methods section-Upon reading the manuscript (rather the results) one discovers that its only rainfall being referred to. Is rainfall enough to define meteorological characteristics? - what about the influence of other climate characteristics such as Temperature, Evapo-transpiration?

Reply: Thanks. (1) The term "climate characteristics" in this manuscript refers to the variables of Y1-Y24 in Table 2, which include rainfall, temperature, potential evapotranspiration, and runoff. It also should be mentioned that the employed climate variables can be divided into four categories, i.e., daily (Y1-Y4), monthly (Y5-Y7), seasonal (Y8-Y16), and annual scale variables (Y17-Y24). Please refer to Table 2. (2) Since rainfall is the most important factor that influences the degree of catchment wetness, the identification of meteorological drought in this manuscript was only based on the annual rainfall data as in other studies (Li et al., 2020; Pan et al., 2019b; Saft et al., 2015; Wong et al., 2013). Furthermore, Saft et al. (2015) indicated that the selected algorithm has been verified as a rigorous method for processing the autocorrelation in regression residuals and testing the global significance. Furthermore, we have the same study region, i.e., catchments in southeastern Australia (but our data sources are different). Thus, the method proposed by Saft et al. (2015) was employed in this study to define the meteorological drought period.

Add reference: Li, Q. F., He, P. F., He, Y. C., Han, X. Y., Zeng, T. S., Lu, G. B., and Wang, H. J.: Investigation to the relation between meteorological drought and hydrological drought in the upper Shaying River Basin using wavelet analysis, Atmos. Res., 234, 10, 10.1016/j.atmosres.2019.104743, 2020. Wong, G., van Lanen, H. A. J., and Torfs, P.: Probabilistic analysis of hydrological drought characteristics using meteorological drought, Hydrol. Sci. J.-J. Sci. Hydrol., 58, 253-270, 10.1080/02626667.2012.753147, 2013.

B5: In section 4.3 authors point out that 'However, the geographical distribution of catchments with significant and non-significant changes in $\theta\_1$ showed no obvious geolocation clustering phenomenon. Which statistical techniques did the authors employ to come up with conclusions on significance in $\theta\_1$ Given multiple climate variables and catchment properties-which one significantly affects CWSC?

Reply: Thanks. (1) In order to improve the clarity, this sentence will be deleted in the modified manuscript. (2) The criteria in identifying catchments with a significant/non-significant change in $\theta\_1$ were presented in section 3.2.5. As illustrated in lines 312-317, the minimum requirement of significant change in $\theta\_1$ was defined as the simulated values of the parameter $\theta\_1$ between the two periods should be more than $\pm20\%$. In other words, only the catchments with more than $\pm20\%$ changes in $\theta\_1$ would be recognized as changed significantly. After a comparison of several other threshold levels (such as $\pm5\%$ and $\pm10\%$), we found that the value of $\pm20\%$ can maximally exclude the negative impacts by the heterogeneity of the available parameter sets. (3) As presented in Figures 7 and 9, we used the Pearson Correlation Coefficient to reflect the potential relation between the (%) shift in the CWSC and the catchment properties/climate characteristics. However, no strong association has been found between the shift in the CWSC and the single climate variable/catchment property. Thus, it is really hard to judge the influence of each factor on the CWSC. According to our study, it seems that the final changes in the CWSC are the combined effects of multiple climate variables and catchment properties. Thus, we expected that further studies are still required to confirm which factors played the most important role in the catchment dynamic.

Please also note the supplement to this comment:
https://www.hydrol-earth-syst-sci-discuss.net/hess-2019-692/hess-2019-692-AC2-supplement.pdf

---

## Author Response (AR1)

Responses to all the Referees:

Dear Editor and Reviewers:

We sincerely appreciate the comments and advice from the Editor and Referees which have not only improved the quality of the current manuscript greatly, but also are beneficial for our research in general. We have carefully followed these comments in making revisions. Our detailed responses to the comments raised by the Editor and Referees are presented below. In the following Responses, for the sake of your evaluation, A1 represents the comment 1 made by Reviewer #1, B1 represents the comment 1 made by Reviewer #2.

Sincerely,
Pan Liu, Ph.D., Professor
School of Water Resources and Hydropower
State Key Laboratory of Water Resources and Hydropower Engineering Science
Wuhan University, Wuhan, Hubei Province, 430072, P. R. China
**E-mail:** liupan@whu.edu.cn

Responses to Editor:
You are kindly asked to individually respond to all referee comments (RCs) that have not yet been answered (marked in red). You can choose between posting a new author comment (AC) and co-listing an existing one in response to an RC. You are also invited to respond to other discussion contributions, if applicable.
**Reply:** We thank the editor for handling the paper, we have carefully followed editor's advice in responding to reviewers' comments. For details, see below.

Responses to Referee #1:
This paper represents an interesting huge work, the main objective of this paper is to detect the possible changes in catchment water storage capacity induced by a prolonged meteorological drought. It will also help improve our ability to simulate the hydrological system. Meanwhile, the structure is clear and analyses and conclusions in this paper are reasonable and logical. Overall, it is a nice paper with proper methodology and complete discussion on the main findings. A minor modification should be considered before publication.

**Reply:** We are grateful for the reviewer's appreciation of our work and for the professional comments, which are carefully followed in making revision.

General comments:
A1: First, the results indicated that 12 catchments had a significant downward shift in the CWSC. My first principal doubts come from there: are 12 sufficient to be able to analyze regularity about catchments with a significant downward shift.

**Reply:** Thanks. We have made it clearer in the revised version that to ensure the accuracy of the modeling results, strict criteria were adopted to evaluate the model results and to identify the catchments with a significant change in $\theta_1$ including the minimum NSE requirement, the minimum requirement of significant change, the requirement for maximum performance degradation, and the requirement for the robustness of results (Please refer to lines 318-336 in the modified manuscript). Only these catchments that satisfied all these criteria were identified as catchments undergo significant changes. Actually, only 12 of the 145 catchments, which had experienced a long-term meteorological drought, satisfied all these four criteria and have been identified with a significant downward shift in the CWSC. Thus, the identified 12 catchments that had a significant downward shift should be reasonable results.

We have also clarified that this result depends on the criteria used for evaluation. All catchments from southeastern Australia in the dataset have been tested in this study using the above-defined criteria, and we did not find more catchments with a significant downward shift in the CWSC. More explanations have been provided in the revision. Please refer to lines 178-187, 318-336, 369-395, and 422-426 in the modified manuscript.

A2: Second: Formulating conclusions is similar to drawing up an abstract. I mean that the emphases of the abstract and conclusions are different.

**Reply:** Thank you for your comments, following which, the Abstract and conclusion have been rewritten.

The part of Abstract has been modified as follows:

Understanding the propagation of prolonged meteorological droughts helps solve the problem of intensified water scarcity around the world. Most of the existing literature studied the propagation of drought from one type to another (e.g., from meteorological to hydrological drought) with statistical approaches, there remains a difficulty in revealing the causality between the meteorological drought and potential changes in the Catchment Water Storage Capacity (CWSC). This study aims to identify the response of the CWSC to the meteorological drought by examining the changes of hydrological model parameters after the drought events. Firstly, the temporal variation of a model parameter that denotes the CWSC is estimated to reflect the potential changes in real CWSC. Next, the change points of the CWSC parameter were determined based on the Bayesian change point analysis. Finally, the possible association and linkage between the shift in the CWSC and the time-lag of the catchment (i.e., time-lag between the onset of the drought and the change point) with multiple catchment properties and climate characteristics were identified. In total 145 catchments from southeastern Australia were selected as the study areas. Results indicated that (1) significant shifts in the CWSC can be observed in 62.7% of the catchments, which can be divided into two subsets with the opposite response, i.e. 48.2% of catchments had lower runoff generation rates while 14.5% of catchments had higher runoff generation rate; (2) the increase in the CWSC during a chronic drought can be observed in smaller catchments with lower elevation, slope, and forest coverage of

Evergreen Broadleaf Forest, while the decrease in the CWSC can be observed in larger catchments with higher elevation and larger coverage of the Evergreen Broadleaf Forest; (3) catchments with a lower proportion of Evergreen Broadleaf Forest usually have longer time-lag and are more resilient. This study improves our understanding of possible changes in the CWSC induced by a prolonged meteorological drought, which will help improve our ability to simulate the hydrological system under climate change.

Please refer to lines 18-40 in the modified manuscript.

The Conclusion has been modified as follows:

This study aims to examine the possible changes in the CWSC as well as the time-lag between the onset of the meteorological drought and the change point of the CWSC. A classical hydrological model, GR4J, was used and its parameter $\theta_1$ was selected to denote catchment water storage capacity (CWSC). Thus, the temporal variation in parameter $\theta_1$ was detected to reveal the possible fluctuation in the CWSC, and the causality between the temporal variation in parameter $\theta_1$ and a persistent meteorological drought was examined. One hundred forty-five catchments in southeastern Australia were selected as the study areas. Main conclusions can be drawn as follows:

(1) Significant changes in the CWSC have been identified in 62.7% (52 in 83) of catchments, which can be divided into two subsets with opposite catchment responses: 48.2% (40 in 83) experienced a significant decrease in the CWSC during the drought period and had lower runoff generation rates, while 14.5% (12 in 83) of catchments experienced a significant decrease in the CWSC during the drought period, and had higher runoff generation rate.

(2) Different change directions in the CWSC resulted in the opposite impacts on runoff generation, i.e., catchments with increased CWSC would result in lower runoff generation rates for similar amounts of rainfall than before while those catchments with decreased CWSC would have an opposite response (higher runoff generation rate). Generally, the increase in the CWSC during a chronic drought can be observed in smaller catchments with lower elevation, slope, and forest coverage of Evergreen Broadleaf Forest, while the decrease in the CWSC can be observed in larger catchments with higher elevation and larger coverage of the Evergreen Broadleaf Forest. Among all catchment properties and climate variables considered, our results suggest that two climate variables (i.e., variation in annual rainfall and annual runoff ratio) have the strongest associations with the shift in the CWSC.

(3) The responses of different catchments to persistent meteorological drought were not equally susceptible. Catchments with a lower proportion of Evergreen Broadleaf Forest usually have longer time-lag and are more resilient.

It is noted that although this study resulted in interesting findings that give new insight and have not been fully outlined before, it is based on the lumped GR4J model and the specific case in Australia, which implies that the main findings/conclusions may not directly extendable to other regions. Thus, to examine the generality of the main conclusions, the response of CWSC to the meteorological drought can be analyzed with the other hydrological models in the other regions.

Please refer to lines 602-632 in the modified manuscript.

Detailed comments:

A3: Is there a mistake in the inconsistency between the Abstract (L37-L38) and Conclusion (L613-615).

**Reply:** We are sorry for this error. The related has been modified as "…the increase in the CWSC during a chronic drought can be observed in smaller catchments with lower elevation, slope, and forest coverage of Evergreen Broadleaf Forest, while the decrease in the CWSC can be observed in larger catchments with higher elevation and larger coverage of the Evergreen Broadleaf Forest.". Please refer to lines 33-37 and 618-621 in the modified manuscript.

A4: Previous study (Yan et al., 2015) has indicated that the sensitivity of the parameters in the GR4J model has different performance in different parts of the same basin. Therefore, you should cite more citations or do some researches in the study area to support the sentence in the L224-L226.

**Reply:** Thank you. More citations have been added to the revised manuscript. This sentence has been modified as "Previous studies (Demirel et al., 2013; Pan et al., 2019a, 2019b; Perrin et al., 2003; Westra et al., 2014; Yan et al., 2015) showed that $\theta_1$, which denotes the catchment water storage capacity, is the most sensitive parameter in the structure of the GR4J model.". Please refer to lines 236-239 in the modified manuscript.

Added reference:

1. Yan, X. L., Zhang, J. Y., Wang, G. Q., Bao, Z. X., Liu, C. S., and Xuan, Y. Q.: Application of GR4J Rainfall-runoff Model to Typical Catchments in the Yellow River Basin, Proceedings of the 5th International Yellow River Forum on Ensuring Water Right of the River's Demand and Healthy River Basin Maintenance, Vol V, Yellow River Conservancy Press, Zhengzhou, 191-198 pp., 2015.

2. Demirel, M. C., Booij, M. J., and Hoekstra, A. Y.: Effect of different uncertainty sources on the skill of 10 day ensemble low flow forecasts for two hydrological models, Water Resour. Res., 49, 4035-4053, 10.1002/wrcr.20294, 2013.

A5: L503-517: You should cut out much of the repetition with the Section 4.

**Reply:** Thanks. Changes have been made as suggested. The paragraph in these lines has been modified as follows:

The results indicate that, under certain circumstances, a long-term meteorological drought would result in a significant change in the CWSC. However, no strong association has been found between the magnitude of the change in the CWSC with any single variable. In addition, the length of dry period was not associated with the shift in the CWSC. Thus, it seems that the catchment response behavior to long-term meteorological drought is controlled by the combination of local catchment properties and climate characteristics rather than a single factor. Thus, further studies are still required to confirm which factors played the most important role in the catchment dynamic.

Please refer to lines 515-521 in the modified manuscript.

A6: In the Section 4.4 You had better discuss the factors for the direction first and then the magnitude of shifts in the CWSC.

**Reply:** Thanks. We have explained better that Section 4.4.1 presents the results that classify the catchments into two groups, one group shows significant shifts in the CWSC, and another does not. While section 4.4.2 continued to discuss the difference between two sub-groups of catchments with significant upward/significant downward shifts in the CWSC. These two sub-groups of catchments were extracted from the group of catchments with significant shifts in the CWSC according to the change direction of the estimated $\theta_1$. Thus, we thought it might be better to discuss the difference between two groups of catchments with/without significant shifts at first, and then illustrate the difference between two subsets of catchments with significant upward/significant downward shifts later.

To make it clearer, after considering reviewer's advice and reminder, the headings of sections 4.4.1 and 4.4.2 have been modified as "4.4.1 Factors for the significant/non-significant shifts in the CWSC" and "4.4.2 Factors for the significant upward/downward shifts in the CWSC" in the revised manuscript.

A7: Why not analyze the reasons for conclusion 2 in the Section 5.1? (Catchments with larger elevation and slope, lower forest coverage of Evergreen Broad leaf Forest are more likely to have an increase in the CWSC during a chronic drought while smaller catchments with lower elevation, lower coverage of the Evergreen Broad leaf Forest are more likely to have a decrease in the CWSC.)

**Reply:** Thanks. We tried but as shown in Figures 7 and 9, no strong PCC association has been found between the magnitude of the change in the CWSC and the single catchment property or climate variable. It seems that the catchment response behavior to long-term meteorological drought is controlled by the combination of local catchment properties and climate characteristics rather than a single factor Thus, we prospected that further studies are still required to confirm which factors played the most important role in the catchment dynamic.

A8: The conclusion should be concise and coherent. It is suggested that only the key and definitive conclusions of this paper should be stated. If you cannot confirm your conclusion, these results may only be used in the Discussion, such as L610-611 and L622-631, etc.

**Reply:** Thanks. Changes have been made as suggested. The sentences in lines 610-611 (in the original manuscript) have been deleted in the conclusion part of the revised manuscript. In addition, the sentences in lines 622-631 (in the original manuscript) have been modified as follows:

It is noted that although this study resulted in interesting findings that give new insight and have not been fully outlined before, it is based on the lumped GR4J model and the specific case in Australia, which implies that the main findings/conclusions may not directly extendable to other regions. Thus, to examine the generality of the main conclusions, the response of CWSC to the meteorological drought can be analyzed with the other hydrological models in the other regions.

Please refer to lines 628-632 in the modified manuscript.

A9: It's better to add latitude and longitude to the location map of the study catchments (Figure 1).
**Reply:** Thanks. Changes have been made in the revised Figure 1. The modified Figure 1 reads as follows.

[Figure]

Figure 1. Location map of the study catchments in southeastern Australia. The dark gray color denotes the catchments with a long-term meteorological drought (145 catchments) while the light gray color denotes the catchments without any sustained droughts or ha more than one prolonged drought period (253 catchments).

A10: Table 3 should add the results about the fourth criteria.
**Reply:** Thanks. Changes have been made in the revised Table 3. The modified Table 3 is as follows.

Table 3. The direction of the shifts in the CWSC due to the long-term meteorological drought for the catchments in southeastern Australia.

| Magnitude | Change direction | Percentage (Number of catchments) |
|---|---|---|
| Significant change | Downward (Smaller CWSC than the previous estimation suggests) | 8.3% (12) |
| | Upward (Larger CWSC than the previous | 27.6% (40) |

| | estimation suggests) | |
|---|---|---|
| Non-significant | Slight increase | 12.4% (18) |
| change | Slight decrease | 9.0% (13) |
| Dissatisfy the criteria of the minimum NSE performance, the maximum performance degradation and the robustness requirement | | 42.8% (62) |
| All (catchments with a sustained meteorological drought) | | 100% (145) |

A11: Figure 4 can be improved. The figure should show the change directions and magnitudes of the CWSC at the same time. It is better for readers to understand the shifts of the CWSC in the adjacent catchments.

**Reply:** Thanks. Changes have been made as suggested. The modified Figure 4 reads as follows:

[Figure]

Figure 4. Location map of the catchments with the significant and non-significant shifts in the CWSC. The red (blue) color denotes the catchments that have a significant increase (decrease) in the CWSC after the Change point while the (light blue) light red color denotes the catchments with a non-significant increase (decrease) in the CWSC.

A12: Is the supplementary material missing?

**Reply:** We are sorry for this mistake. We have uploaded the supplementary material (Table S1) to the system. Please also refer to the material (Table S1) below.

Table S1. Catchments with the long-term meteorological droughts.

[revised manuscript text omitted]

Responses to Referee #2:

I read with interest the paper on The Influence of a Prolonged Meteorological Drought on the Catchment 1 Water Storage Capacity: A Hydrological Model Perspective. It's a good piece of work whose findings are useful for water resources management in the light of climate and landuse changes. My major concern however is in the study design and methodology. Most of the steps in the methods section are not adequately described. Readers can fully link the methods with the results because certain information is not provided in the methods section.

**Reply:** Thanks for the comment and suggestions, we have carefully improved the description of the study design and methodology part in the revision. Some examples are described below:

B1: Typical examples but not limited to: Which datasets have been used in all the objectives Describe the spatial and temporal variation of these datasets What are the different sources of datasets For examples what is the source of the primary land-use types described in the results section.

**Reply:** Thanks. More descriptions of the adopted data set have been added in the revised manuscript. The sentences in lines 159-166 in the original manuscript for describing the research data has been modified as follows:

The following data have been used in this study: (1) climate variables, include daily rainfall and daily potential evapotranspiration; (2) daily streamflow observation at catchment outlet; (3) land use types at 1 km resolution; (4) soil types at 30 arc-second resolution, and (5) catchment attributes, include catchment area, mean elevation and so on. The detailed lists of the catchment attributes and climate characteristics were presented in Table 2. The data of climate variables, daily runoff, and catchment attributes were obtained from the Australian Water Resources Assessment (AWRA) system, which has been served as a standard publicly available national dataset for hydrological                    model                    evaluation
(https://publications.csiro.au/rpr/pub?pid=csiro:EP113194, Zhang et al. (2013)). For all
catchments, there is no missing data in the rainfall and potential evapotranspiration data
while the runoff data in some catchments are missing. The data set of soil types was
obtained from the Harmonized World Soil Database by the Food and Agriculture
Organization of the United Nations (http://www.fao.org/soils-portal/soil-survey/soil-
maps-and-databases/harmonized-world-soil-database-v12/en/, Fischer et al. (2008))
and was classified according to the Soil Texture Triangle of USDA
(https://www.nrcs.usda.gov/wps/portal/nrcs/detail/soils/survey/?cid=nrcs142p2_0541
67). The data set of the land use types was derived from the global land cover map
released by the University of Maryland (UMD) (Hansen et al., 2000) and was classified
according to the UMD Land Cover Classification method (http://app.earth-
observer.org/data/basemaps/images/global/LandCover_512/LandCoverUMD_512/La
ndCoverUMD_512.html).

Please refer to lines 158-177 in the modified manuscript.

B2: Authors mention about the catchment physical properties and climate
characteristics which influences CWSC. Readers will only get to understand or know
these physical properties and climate characteristics when they read the results section.
This again has to do with a poor study design.
**Reply:** Thanks. The list of catchment properties and climate characteristics that might
influence CWSC was presented in Table 2. And the illustrations of these potential
factors were presented in section 3.4. To facilitate the understanding of this manuscript,
the following sentence has been added in section 2.2. of the modified manuscript:

The detailed lists of the catchment attributes and climate characteristics were
presented in Table 2.

Please refer to lines 161-162 in the modified manuscript.

B3: In section 3.4, what do authors mean by '. . ...because of the limitation of available
data of catchment attributes, only unique catchment properties are employed. . .". What
are these unique catchment properties? And how were they selected?
**Reply:** Thanks.

(1) The 'unique catchment properties' we meant these data are static values rather
than a time series. For example, only one value of the catchment area was used.

(2) The unique catchment properties refer to the X1-X9 in Table 2, i.e., catchment
area, mean elevation and elevation difference between the maximum and minimum
elevations, mean slope, and slope range, forest coverage percentage, saturated hydraulic
conductivity of the top soil, available soil water holding capacity of the top soil and sub
soil.

(3) These catchment attributes were selected because they reflect the physical
characteristics of the catchment, and might be related to the shift in the CWSC.

(4) To make it clearer, this sentence have been modified as "It should be noted that
due to the limitation of available data of catchment attributes, for each catchment, only
one static/constant value of the catchment property was employed (X1-X9).".  Please refer to lines 354-356 in the modified manuscript.

B4: The authors use the term 'climate characteristics' in the abstract and methods section-Upon reading the manuscript (rather the results) one discovers that its only rainfall being referred to. Is rainfall enough to define meteorological characteristics? - what about the influence of other climate characteristics such as Temperature, Evapotranspiration?
**Reply:** Thanks.

(1) The term "climate characteristics" in this manuscript refers to the variables of Y1-Y24 in Table 2, which include rainfall, temperature, potential evapotranspiration, and runoff. It also should be mentioned that the employed climate variables can be divided into four categories, i.e., daily (Y1-Y4), monthly (Y5-Y7), seasonal (Y8-Y16), and annual scale variables (Y17-Y24). Please refer to Table 2.

(2) Since rainfall is the most important factor that influences the degree of catchment wetness, the identification of meteorological drought in this manuscript was only based on the annual rainfall data as in other studies (Li et al., 2020; Pan et al., 2019b; Saft et al., 2015; Wong et al., 2013). Furthermore, Saft et al. (2015) indicated that the selected algorithm has been verified as a rigorous method for processing the autocorrelation in regression residuals and testing the global significance. Furthermore, we have the same study region, i.e., catchments in southeastern Australia (but our data sources are different). Thus, the method proposed by Saft et al. (2015) was employed in this study to define the meteorological drought period.

Add reference:
1. Li, Q. F., He, P. F., He, Y. C., Han, X. Y., Zeng, T. S., Lu, G. B., and Wang, H. J.: Investigation to the relation between meteorological drought and hydrological drought in the upper Shaying River Basin using wavelet analysis, Atmos. Res., 234, 10, 10.1016/j.atmosres.2019.104743, 2020.
2. Wong, G., van Lanen, H. A. J., and Torfs, P.: Probabilistic analysis of hydrological drought characteristics using meteorological drought, Hydrol. Sci. J.-J. Sci. Hydrol., 58, 253-270, 10.1080/02626667.2012.753147, 2013.

B5: In section 4.3 authors point out that 'However, the geographical distribution of catchments with significant and non-significant changes in $\theta_1$ showed no obvious geolocation clustering phenomenon. Which statistical techniques did the authors employ to come up with conclusions on significance in $\theta_1$ Given multiple climate variables and catchment properties-which one significantly affects CWSC?
**Reply:** Thanks.

(1) In order to avoid unclearness, this sentence has been deleted in the modified manuscript.

(2) The criteria in identifying catchments with a significant/non-significant change in $\theta_1$ were presented in section 3.2.5. As illustrated in lines 325-330, the minimum requirement of significant change in $\theta_1$ was defined as the simulated values of the parameter $\theta_1$ between the two periods should be more than ±20%. In other words, only the catchments with more than ±20% changes in $\theta_1$ would be recognized as changed significantly. After a comparison of several other threshold levels (such as ±5% and ±10%), we found that the value of ±20% can maximally exclude the negative impacts by the heterogeneity of the available parameter sets.

(3) As presented in Figures 7 and 9, we used the Pearson Correlation Coefficient to reflect the potential relation 
[revised manuscript text omitted]